# Model-Based Offline Reinforcement Learning with Pessimism-Modulated Dynamics Belief

**Kaiyang Guo**[*]     **Yunfeng Shao**     **Yanhui Geng**
Huawei Noah's Ark Lab

## Abstract

Model-based offline reinforcement learning (RL) aims to find highly rewarding policy, by leveraging a previously collected static dataset and a dynamics model. While the dynamics model learned through reuse of the static dataset, its generalization ability hopefully promotes policy learning if properly utilized. To that end, several works propose to quantify the uncertainty of predicted dynamics, and explicitly apply it to penalize reward. However, as the dynamics and the reward are intrinsically different factors in context of MDP, characterizing the impact of dynamics uncertainty through reward penalty may incur unexpected tradeoff between model utilization and risk avoidance. In this work, we instead maintain a belief distribution over dynamics, and evaluate/optimize policy through biased sampling from the belief. The sampling procedure, biased towards pessimism, is derived based on an alternating Markov game formulation of offline RL. We formally show that the biased sampling naturally induces an updated dynamics belief with policy-dependent reweighting factor, termed *Pessimism-Modulated Dynamics Belief*. To improve policy, we devise an iterative regularized policy optimization algorithm for the game, with guarantee of monotonous improvement under certain condition. To make practical, we further devise an offline RL algorithm to approximately find the solution. Empirical results show that the proposed approach achieves state-of-the-art performance on a wide range of benchmark tasks.

## 1 Introduction

In typical paradigm of RL, the agent actively interacts with environment and receives feedback to promote policy improvement. The essential trial-and-error procedure can be costly, unsafe or even prohibitory in practice (e.g. robotics [1], autonomous driving [2], and healthcare [3]), thus constituting a major impediment to actual deployment of RL. Meanwhile, for a number of applications, historical data records are available to reflect the system feedback under a predefined policy. This raises the opportunity to learn policy in purely offline setting.

In offline setting, as no further interaction with environment is permitted, the dataset provides a limited coverage in state-action space. Then, the policy that induces out-of-distribution (OOD) state-action pairs can not be well evaluated in offline learning phase, and deploying it online potentially attains terrible performance. Recent studies have reported that applying vanilla RL algorithms to offline dataset exacerbates such a distributional shift [4–6], making them unsuitable for offline setting.

To tackle the distributional shift issue, a number of offline RL approaches have been developed. Specifically, one category of them propose to directly constrain the policy close to the one collecting data [4, 5, 7, 8], or penalize Q-value towards conservatism for OOD state-action pairs [9–11]. While they achieve remarkable performance gains, the policy regularizer and the Q-value penalty tightly restricts the produced policy within the data manifold. Instead, more recent works consider to

---

[*]Corresponding to: `guokaiyang@huawei.com`

36th Conference on Neural Information Processing Systems (NeurIPS 2022).

quantify the uncertainty of Q-value with neural network ensembles [12], where the consistent Q-value estimates indicate high confidence and can be plausibly used during learning process, even for OOD state-action pairs [13, 14]. However, the uncertainty quantification over OOD region highly relies on how neural network generalizes [15]. As the prior knowledge of Q-function is hard to acquire and insert into the neural network, the generalization is unlikely reliable to facilitate meaningful uncertainty quantification [16]. Notably, all these works are model-free.

Model-based offline RL optimizes policy based on a constructed dynamics model. Compared to the model-free approaches, one prominent advantage is that the prior knowledge of dynamics is easier to access. First, the generic prior like smoothness widely exists in various domains [17]. Second, the sufficiently learned dynamics models for relevant tasks can act as a data-driven prior for the concerned task [18–20]. With richer prior knowledge, the uncertainty quantification for dynamics is more trustworthy. Similar to the model-free approach, the dynamics uncertainty can be incorporated to find reliable policy beyond data coverage. However, an additional challenge is how to characterize the accumulative impact of dynamics uncertainty on the long-term reward, as the system dynamics is with entirely different meaning compared to the reward or Q-value.

Although existing model-based offline RL literature theoretically bounds the impact of dynamics uncertainty on final performance, their practical variants characterize the impact through reward penalty [6, 21, 22]. Concretely, the reward function is penalized by the dynamics uncertainty for each state-action pair [21], or the agent is enforced to a low-reward absorbing state when the dynamics uncertainty exceeds a certain level [6]. While optimizing policy in these constructed MDPs stimulates anti-uncertainty behavior, the final policy tends to be over-conservative. For example, even the transition dynamics for a state-action pair is ambiguous among several possible candidates, these candidates may generate the states from which the system evolves similarly.[2] Then, such a state-action pair should not be treated specially.

Motivated by the above intuition, we propose pessimism-modulated dynamics belief for model-based offline RL. In contrast with the previous approaches, the dynamics uncertainty is not explicitly quantified. To characterize its impact, we maintain a belief distribution over system dynamics, and the policy is evaluated/optimized through biased sampling from it. The sampling procedure, biased towards pessimism, is derived based on an alternating Markov game (AMG) formulation of offline RL. We formally show that the biased sampling naturally induces an updated dynamics belief with policy-dependent reweighting factor, termed *Pessimism-Modulated Dynamics Belief*. Besides, the degree of pessimism is monotonously determined by the hyperparameters in sampling procedure.

The considered AMG formulation can be regarded as a generalization of robust MDP, which is proposed as a surrogate to optimize the percentile performance in face of dynamics uncertainty [23, 24]. However, robust MDP suffers from two significant shortcomings: 1) The percentile criterion is over-conservative since it fixates on a single pessimistic dynamics instance [25, 26]; 2) Robust MDP is constructed based on an uncertainty set, and the improper choice of uncertainty set would further aggravate the degree of conservatism [27, 28]. The AMG formulation is kept from these shortcomings. To solve the AMG, we devise an iterative regularized policy optimization algorithm, with guarantee of monotonous improvement under certain condition. To make practical, we further derive an offline RL algorithm to approximately find the solution, and empirically evaluate it on the offline RL benchmark D4RL. The results show that the proposed approach obviously outperforms previous state-of-the-art (SoTA) in 9 out of 18 environment-dataset configurations and performs competitively in the rest, without tuning hyperparameters for each task. The proof of theorems in this paper are presented in Appendix B.

## 2 Preliminaries

**Markov Decision Process (MDP)**    A MDP is depicted by the tuple $(\mathcal{S}, \mathcal{A}, T, r, \rho_0, \gamma)$, where $\mathcal{S}, \mathcal{A}$ are state and action spaces, $T(s'|s, a)$ is the transition probability, $r(s, a)$ is the reward function, $\rho_0(s)$ is the initial state distribution, and $\gamma$ is the discount factor. The goal of RL is to find the policy $\pi : s \rightarrow \Delta(\mathcal{A})$ that maximizes the cumulative discounted reward:

$$J(\pi, T) = \mathbb{E}_{\rho_0, T, \pi}\left[\sum_{t=0}^{\infty} \gamma^t r(s_t, a_t)\right], \tag{1}$$

---

[2]Or from these states, the system evolves differently but generates similar rewards.

where $\Delta(\cdot)$ denotes the probability simplex. In typical RL paradigm, this is done via actively interacting with environment.

**Offline RL**  In offline setting, the environment is unaccessible, and only a static dataset $\mathcal{D} = \{(s, a, r, s')\}$ is provided, containing the previously logged data samples under an unknown behavior policy. Offline RL aims to optimize the policy by solely leveraging the offline dataset.

To simplify the presentation, we assume the reward function $r$ and initial state distribution $\rho_0$ are known. Then, the system dynamics is unknown only in terms of the transition probability $T$. Note that the considered formulation and the proposed approach can be easily extend to the general case without additional technical modification.

**Robust MDP**  With the offline dataset, a straightforward strategy is first learning a dynamics model $\tau(s'|s, a)$ and then optimizing policy via simulation. However, due to the limitedness of available data, the learned model is inevitably imprecise. Robust MDP [23] is a surrogate to optimize policy with consideration of the ambiguity of dynamics. Concretely, robust MDP is constructed by introducing an uncertainty set $\mathcal{T} = \{\tau\}$ to contain plausible transition probabilities. If the uncertainty set includes the true transition with probability of $(1 - \delta)$, the performance of any policy $\pi$ in true MDP can be lower bounded by $\min_{\tau \in \mathcal{T}} J(\pi, \tau)$ with probability of at least $(1 - \delta)$. Thus, the percentile performance for the true MDP can be optimized by finding a solution to

$$\max_\pi \min_{\tau \in \mathcal{T}} J(\pi, \tau). \tag{2}$$

Despite its popularity, Robust MDP suffers from two major shortcomings: First, the percentile criterion overly fixates on a single pessimistic transition instance, especially when there are multiple optimal policies for this transition but they lead to dramatically different performance for other transitions [25, 26]. This behavior results in unnecessarily conservative policy.

Second, the level of conservatism can be further aggravated when the uncertainty set is inappropriately constructed [27]. For a given policy $\pi$, the ideal situation is that $\mathcal{T}$ contains the $(1 - \delta)$ proportion of transitions with which the policy achieves higher performance than with the other $\delta$ proportion. Then, $\min_{\tau \in \mathcal{T}} J(\pi, \tau)$ is exactly the $\delta$-quantile performance. This requires the uncertainty set to be policy-dependent, and during policy optimization the uncertainty set should change accordingly. Otherwise, if $\mathcal{T}$ is predetermined and fixed, it is possible to have $\tau' \notin \mathcal{T}$ with non-zero probability and satisfying $J(\pi^*, \tau') > \min_{\tau \in \mathcal{T}} J(\pi^*, \tau)$, where $\pi^*$ is the optimal policy for (2). Then, adding $\tau'$ into $\mathcal{T}$ does not affect the optimal solution of the problem (2). This indicates that we are essentially optimizing a $\delta'$-quantile performance, where $\delta'$ can be much smaller than $\delta$. In literature, the uncertainty sets are mostly predetermined before policy optimization [23, 29–31].

## 3  Pessimism-Modulated Dynamics Belief

In short, robust MDP is over-conservative due to the fixation on a single pessimistic transition instance and the predetermination of uncertainty set. In this work, we strive to take the entire spectrum of plausible transitions into account, and let the algorithm by itself determine which part deserves more attention. To this end, we consider an alternating Markov game formulation of offline RL, based on which the proposed offline RL approach is derived.

### 3.1  Formulation

**Alternating Markov game (AMG)**  The AMG is a specialization of two-player zero-sum game, depicted by $(\mathcal{S}, \bar{\mathcal{S}}, \mathcal{A}, \bar{\mathcal{A}}, G, r, \rho_0, \gamma)$. The game starts from a state sampled from $\rho_0$, then two players alternatively choose actions $a \in \mathcal{A}$ and $\bar{a} \in \bar{\mathcal{A}}$ under states $s \in \mathcal{S}$ and $\bar{s} \in \bar{\mathcal{S}}$, along with the game transition defined by $G(\bar{s}|s, a)$ and $G(s|\bar{s}, \bar{a})$. At each round, the primary player receives reward $r(s, a)$ and the secondary player receives its negative counterpart $-r(s, a)$.

**Offline RL as AMG**  We formulate the offline RL problem as an AMG, where the primary player optimizes a reliable policy for our concerned MDP in face of stochastic disturbance from the secondary player. The AMG is constructed by augmenting the original MDP. As both have the transition probability, we use game transition and system transition to differentiate them.

For the primary player, its state space $\mathcal{S}$, action space $\mathcal{A}$ and reward function $r(s,a)$ are same with those in the original MDP. After the primary player acts, the game emits a $N$-size set of system transition candidates $\mathcal{T}^{sa}$, which later acts as the state of secondary player. Formally, $\mathcal{T}^{sa}$ is generated according to

$$G\left(\bar{s} = \mathcal{T}^{sa} | s, a\right) = \prod_{\tau^{sa} \in \mathcal{T}^{sa}} \mathbb{P}_T^{sa}(\tau^{sa}), \tag{3}$$

where $\tau^{sa}(\cdot)$ re-denotes the plausible system transition $\tau(\cdot | s, a)$ for short, and $\mathbb{P}_T^{sa}$ is a given belief distribution over $\tau^{sa}$. According to (3), the elements in $\mathcal{T}^{sa}$ are independent and identically distributed samples following $\mathbb{P}_T^{sa}$. The major difference to uncertainty set in robust MDP is that the set introduced here is unfixed and stochastic for each step. To distinguish with uncertainty set, we call it candidate set. The belief distribution $\mathbb{P}_T^{sa}$ can be chosen arbitrarily to incorporate knowledge of system transition. Particularly, when the prior distribution of system transition is accessible, $\mathbb{P}_T^{sa}$ can be obtained as the posterior by integrating the prior and the evidence $\mathcal{D}$ through Bayes' rule.

The secondary player receives the candidate set $\mathcal{T}^{sa}$ as state. Thus, its state space can be denoted by $\bar{\mathcal{S}} = \Delta^N(\mathcal{S})$, i.e., the n-fold Cartesian product of probability simplex over $\mathcal{S}$. Note that the state $\mathcal{T}^{sa}$ also takes the role of action space, i.e., $\bar{\mathcal{A}} = \mathcal{T}^{sa}$, meaning that the action of secondary player is to choose a system transition from the candidate set. Given the chosen $\tau^{sa} \in \mathcal{T}^{sa}$, the game evolves by sampling $\tau^{sa}$, i.e.,

$$G\left(s' | \bar{s} = \mathcal{T}^{sa}, \bar{a} = \tau^{sa}\right) = \tau^{sa}(s'), \tag{4}$$

and the primary player receives $s'$ to continue the game. In the following, we use $\mathbb{P}_T^N(\mathcal{T}^{sa})$ to compactly denote the game transition $G\left(\bar{s} = \mathcal{T}^{sa} | s, a\right)$ in (3), and omit the superscript $sa$ in $\tau^{sa}$, $\mathcal{T}^{sa}$ and $\mathbb{P}_T^{sa}$ when it is clear from the context.

For the above AMG, we consider a specific policy (explained below) for the secondary player, such that the cumulative discounted reward of the primary player with policy $\pi$ can be written as:

$$J(\pi) := \mathop{\mathbb{E}}_{\rho_0, \pi, \mathbb{P}_T^N} \lfloor \min \rfloor_{\tau_0 \in \mathcal{T}_0}^k \left[ \mathop{\mathbb{E}}_{\tau_0, \pi, \mathbb{P}_T^N} \lfloor \min \rfloor_{\tau_1 \in \mathcal{T}_1}^k \cdots \left[ \mathop{\mathbb{E}}_{\tau_\infty, \pi} \left[ \sum_{t=0}^\infty \gamma^t r(s_t, a_t) \right] \right] \right], \tag{5}$$

where the subscripts of $\tau$ and $\mathcal{T}$ denote time step, the expectation is over $s_0 \sim \rho_0$, $s_{t>0} \sim \tau_{t-1}(\cdot | s_{t-1}, a_{t-1})$, $a_t \sim \pi(\cdot | s_t)$ and $\mathcal{T}_t \sim \mathbb{P}_T^N$, and the operator $\lfloor \min \rfloor_{x \in \mathcal{X}}^k f(x)$ denotes finding $k$th minimum of $f(x)$ over $x \in \mathcal{X}$. The policy of secondary player is implicitly defined by the operator $\lfloor \min \rfloor_{x \in \mathcal{X}}^k f(x)$. When changing $k \in \{1, 2, \cdots, N\}$, the secondary player exhibits various degree of adversarial or aggressive disturbance to the future reward. From the view of original MDP, this behavior raises flexible tendency ranging from pessimism to optimism when evaluating policy $\pi$.

The distinctions between the introduced AMG and the robust MDP are twofold: 1) With a belief distribution over transitions, robust MDP will select only part of its supports into uncertainty set, and the set elements are treated indiscriminatingly. It indicates that both the possibility of transitions out of the uncertainty set and the relative likelihood of transitions within the uncertainty set are discarded. However, in the AMG, the candidate set simply contains samples drawn from the belief distribution, implying no information drop in an average sense. Intuitively, by keeping richer knowledge of the system, the performance evaluation is more exact and away from excessive conservatism; 2) In robust MDP, the level of conservatism is expected to be controlled by its hyperparameter $\delta$. However, as illustrated in Section 2, a smaller $\delta$ does not necessarily corresponds to a more conservative performance evaluation, due to the extra impact from uncertainty set construction. In contrast, for the AMG, the degree of conservatism is adjusted by the size of candidate size $N$ and the order of minimum $k$. When changing values of $k$ or $N$, the impact to performance evaluation is ascertained, as formalized in Theorem 3.

To evaluate $J(\pi)$, we define the following Bellman backup operator:

$$\mathcal{B}_{N,k}^\pi Q(s,a) = r(s,a) + \gamma \mathbb{E}_{\mathbb{P}_T^N} \left[ \lfloor \min \rfloor_{\tau \in \mathcal{T}}^k \mathbb{E}_{\tau, \pi} \left[ Q(s', a') \right] \right]. \tag{6}$$

As the operator depends on $N$, $k$ and we emphasize pessimism in offline RL, we call it $(N, k)$-pessimistic Bellman backup operator. Compared to the standard Bellman backup operator in Q-learning, $\mathcal{B}_{N,k}^\pi$ additionally includes the expectation over $\mathcal{T} \sim \mathbb{P}_T^N$ and the $k$-minimum operator over $\mathcal{T}$. Despite these differences, we prove that $\mathcal{B}_{N,k}^\pi$ is still a contraction mapping, based on which $J(\pi)$ can be easily evaluated.

**Theorem 1** (Policy Evaluation). *The $(N,k)$-pessimistic Bellman backup operator $\mathcal{B}_{N,k}^\pi$ is a contraction mapping. By starting from any function $Q : \mathcal{S} \times \mathcal{A} \to \mathbb{R}$ and repeatedly applying $\mathcal{B}_{N,k}^\pi$, the sequence converges to $Q_{N,k}^\pi$, with which we have $J(\pi) = \mathbb{E}_{\rho_0,\pi}\left[Q_{N,k}^\pi(s_0, a_0)\right]$.*

## 3.2 Pessimism-Modulated Dynamics Belief

With the converged Q-value, we are ready to establish a more direct connection between the AMG and the original MDP. The connection appears as the answer to a natural question: the calculation of (6) encompasses biased samples from the dynamics belief distribution, can we treat these samples as the unbiased ones sampling from another belief distribution? We give positive answer in the following theorem.

**Theorem 2** (Equivalent MDP with Pessimism-Modulated Dynamics Belief). *The alternating Markov game in (5) is equivalent to the MDP with tuple $(\mathcal{S}, \mathcal{A}, \widetilde{T}, r, \rho_0, \gamma)$, where the transition probability $\widetilde{T}(s'|s,a) = \mathbb{E}_{\widetilde{\mathbb{P}}_T^{sa}}\left[\tau^{sa}(s')\right]$ is defined with the reweighted belief distribution $\widetilde{\mathbb{P}}_T^{sa}$:*

$$\widetilde{\mathbb{P}}_T^{sa}(\tau^{sa}) \propto w\left(\mathbb{E}_{\tau^{sa},\pi}\left[Q_{N,k}^\pi(s',a')\right]; k, N\right)\mathbb{P}_T^{sa}(\tau^{sa}), \tag{7}$$

$$w(x; k, N) = \left[F(x)\right]^{k-1}\left[1 - F(x)\right]^{N-k}, \tag{8}$$

*and $F(\cdot)$ is cumulative density function. Furthermore, the value of $w(x; k, N)$ first increases and then decreases with $x$, and its maximum is obtained at the $\frac{k-1}{N-1}$ quantile, i.e., $x^* = F^{-1}\left(\frac{k-1}{N-1}\right)$.*

In right-hand side of (7), $\tau^{sa}$ itself is random following the belief distribution, thus $\mathbb{E}_{\tau^{sa},\pi}\left[Q_{N,k}^\pi(s',a')\right]$, as a functional of $\tau^{sa}$, is also a random variable, whose cumulative density function is determined by the belief distribution $\mathbb{P}_T^{sa}$. Intuitively, we can treat $\mathbb{E}_{\tau^{sa},\pi}\left[Q_{N,k}^\pi(s',a')\right]$ as a pessimism indicator for transition $\tau^{sa}$, with larger value indicating less pessimism.

From Theorem 2, the maximum of $w$ is obtained at $\tau^*$: $F\left(\mathbb{E}_{\tau^*,\pi}\left[Q_{N,k}^\pi(s',a')\right]\right) = \frac{k-1}{N-1}$, i.e., the transition with $\frac{k-1}{N-1}$-quantile pessimism indicator. Besides, when $\mathbb{E}_{\tau^{sa},\pi}\left[Q_{N,k}^\pi(s',a')\right]$ departs the $\frac{k-1}{N-1}$ quantile, the reweighting coefficient for its $\tau^{sa}$ decreases. Considering the effect of $w$ to $\widetilde{\mathbb{P}}_T^{sa}$ and the equivalence between the AMG and the refined MDP, we can say that $J(\pi)$ is a soft percentile performance. Compared to the standard percentile criteria, $J(\pi)$ is derived by reshaping belief distribution towards concentrating around a certain percentile, rather than fixating on a single percentile point. Due to this feature, we term $\widetilde{\mathbb{P}}_T^{sa}$ *Pessimism-Modulated Dynamics Belief (PMDB)*.

Lastly, recall that all the above derivations are with hyperparameters $k$ and $N$, we present the monotonicity of $Q_{N,k}^\pi$ over them in Theorem 3. Furthermore, by combining Theorem 1 with Theorem 3, we conclude that $J(\pi)$ decreases with $N$ and increases with $k$.

**Theorem 3** (Monotonicity). *The converged Q-function $Q_{N,k}^\pi$ are with the following properties:*

- *Given any $k$, the Q-function $Q_{N,k}^\pi$ element-wisely decreases with $N \in \{k, k+1, \cdots\}$.*
- *Given any $N$, the Q-function $Q_{N,k}^\pi$ element-wisely increases with $k \in \{1, 2, \cdots, N\}$.*
- *The Q-function $Q_{N,N}^\pi$ element-wisely increases with $N$.*

**Remark 1** (Special Cases). *For $N = k = 1$, we have $\widetilde{\mathbb{P}}_T^{sa} = \mathbb{P}_T^{sa}$. Then, the performance is evaluated through sampling the initial belief distribution. This resembles the common methodology in model-based RL (MBRL), with dynamics belief defined by the uniform distribution over dynamics model ensembles. For $k = \delta(N-1) + 1$ and $N \to \infty$, $\widetilde{\mathbb{P}}_T^{sa}$ asymptotically collapses to be a delta function. Then, $J(\pi)$ degrades to fixate on a single transition instance. It is equivalent to the robust MDP with the uncertainty set constructed as $\left\{\tau^{sa} : \mathbb{P}_T^{sa}(\tau^{sa}) > 0, \mathbb{E}_{\tau^{sa},\pi}\left[Q_{N,k}^\pi(s',a')\right] \geq F^{-1}(\delta)\right\}$. In this sense, the AMG is a successive interpolation between MBRL and robust MDP.*

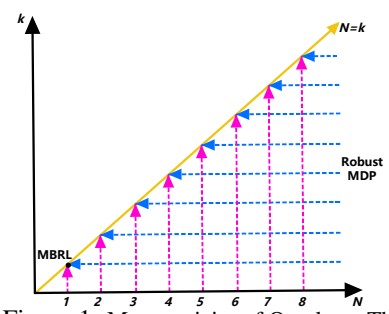

Figure 1: Monotonicity of Q-values. The arrows indicate the directions along which Q-values increase.

# 4  Policy Optimization with Pessimism-Modulated Dynamics Belief

In this section, we optimize policy by maximizing $J(\pi)$. The major consideration is that the methodology should adapt well for both discrete and continuous action spaces. In continuous setting of MDP, the policy can be updated by following stochastic/deterministic policy gradient [32, 33]. However, for the AMG, evaluating $J(\pi)$ itself involves an inner dynamic programming procedure as in Theorem 1. As each evaluation of $J(\pi)$ can only produce one exact gradient, it is inefficient to maximize $J(\pi)$ via gradient-based method. In this section, we consider a series of sub-problems with Kullback–Leibler (KL) regularization. Solving each sub-problem makes prominent update to the policy, and the sequence of solutions for sub-problems monotonously improve regarding $J(\pi)$. Based on this idea, we further derive offline RL algorithm to approximately find the solution.

## 4.1  Iterative Regularized Policy Optimization

Define the KL-regularized return for the AMG by

$$
\bar{J}(\pi; \mu) := \mathop{\mathbb{E}}_{\rho_0, \pi, \mathbb{P}_T^N} \lfloor \min \rfloor_{\tau_0 \in \mathcal{T}_0}^k \left[ \mathop{\mathbb{E}}_{\tau_0, \pi, \mathbb{P}_T^N} \lfloor \min \rfloor_{\tau_1 \in \mathcal{T}_1}^k \cdots \left[ \mathop{\mathbb{E}}_{\tau_\infty, \pi} \left[ \sum_{t=0}^\infty \gamma^t \bigg( r(s_t, a_t) \right. \right. \right.
$$
$$
\left. \left. \left. - \alpha D_{\mathrm{KL}}\big( \pi(\cdot|s_t) \,\|\, \mu(\cdot|s_t) \big) \bigg) \right] \right] \right], \qquad (9)
$$

where $\alpha \geq 0$ is the strength of regularization, and $\mu$ is a reference policy to keep close with.

KL-regularized MDP is considered in previous works to enhance exploration, improve robustness to noise or insert expert knowledge [34–38]. Here, the idea is to constrain the optimized policy in neighbour of a reference policy so that the inner problem is adequately evaluated for such a small policy region.

To optimize $\bar{J}(\pi; \mu)$, we introduce the soft $(N, k)$-pessimistic Bellman backup operator:

$$
\bar{\mathcal{B}}_{N,k}^* Q(s, a) = r(s, a) + \gamma \mathbb{E}_{\mathbb{P}_T^N} \left[ \lfloor \min \rfloor_{\tau \in \mathcal{T}}^k \mathbb{E}_\tau \left[ \alpha \log \mathbb{E}_\mu \exp \left( \frac{1}{\alpha} Q(s', a') \right) \right] \right]. \qquad (10)
$$

**Theorem 4** (Regularized Policy Optimization). *The soft $(N, k)$-pessimistic Bellman backup operator $\bar{\mathcal{B}}_{N,k}^*$ is a contraction mapping. By starting from any function $Q : \mathcal{S} \times \mathcal{A} \to \mathbb{R}$ and repeatedly applying $\bar{\mathcal{B}}_{N,k}^*$, the sequence converges to $\bar{Q}_{N,k}^*$, with which the optimal policy for $\bar{J}(\pi; \mu)$ is obtained as $\bar{\pi}^*(a|s) \propto \mu(a|s) \exp \left( \frac{1}{\alpha} \bar{Q}_{N,k}^*(s, a) \right)$.*

Apparently, the solved policy $\bar{\pi}^*$ depends on the reference policy $\mu$, and setting $\mu$ arbitrarily aforehand can result in suboptimal policy for $J(\pi)$. In fact, we can construct a sequence of sub-problems, with $\mu$ chosen as an improved policy from last sub-problem. By successively solving them, the impact from the initial reference policy is gradually eliminated.

**Theorem 5** (Iterative Regularized Policy Optimization). *By starting from any stochastic policy $\pi_0 : s \to \Delta(\mathcal{A})$ and repeatedly finding $\pi_{i+1} : \bar{J}(\pi_{i+1}; \pi_i) > \bar{J}(\pi_i; \pi_i)$, the sequence of $\{\pi_i\}$ monotonically improves regarding $J(\pi)$, i.e., $J(\pi_{i+1}) \geq J(\pi_i)$. Especially, when $\pi_0(a|s) > 0, \forall s, a$ and $\{\pi_i\}$ are obtained via regularized policy optimization in Theorem 4, we have $\frac{\pi_i(a|s)}{\pi_i(a'|s)} \to \infty$ for any $s, a, a'$ such that $\lim_{i \to \infty} Q_{N,k}^{\pi_i}(s, a) > \lim_{i \to \infty} Q_{N,k}^{\pi_i}(s, a')$.*

Ideally, by combining Theorems 4 and 5, the policy for $J(\pi)$ can be continuously improved by infinitely applying soft pessimistic Bellman backup operator for each of the sequential sub-problems.

**Remark 2** (Iterative Regularized Policy Optimization as Expectation–Maximization with Structured Variational Posterior). *According to Theorem 2, PMDB can be recovered with the converged Q-function $Q_{N,k}^{\pi^*}$. From an end-to-end view, we have an initial dynamics belief $\mathbb{P}_T^{sa}$, then via the calculation based on the belief samples and the reward function, we obtain the updated dynamics belief $\widetilde{\mathbb{P}}_T^{sa}$. It is likely that we are doing some form of posterior inference, where the evidence comes from the reward function. In fact, the iterative regularized policy optimization can be formally recast as an Expectation-Maximization algorithm for offline policy optimization, where the Expectation step correponds to a structured variational inference procedure for dynamics. We elaborate it in Appendix C.*

## 4.2 Offline Reinforcement Learning with Pessimism-Modulated Dynamics Belief

While solving each sub-problem makes prominent update to policy compared with the policy gradient method, we may need to construct several sub-problems before convergence, then exactly solving each of them incurs unnecessary computation. For practical consideration, next we introduce a smooth-evolving reference policy, with which the explicit boundary between sub-problems is blurred. Based on this reference policy, and by further adopting function approximator, we devise an offline RL algorithm to approximately maximize $J(\pi)$.

The idea of smooth-evolving reference policy is inspired by the soft-updated target network in deep RL literature [39, 40]. That is setting the reference policy as a slowly tracked copy of the policy being optimized. Formally, consider a parameterized policy $\pi_\phi$ with the parameter $\phi$. The reference policy is set as $\mu = \pi_{\phi'}$, where $\phi'$ is the moving average of $\phi : \phi' \leftarrow \omega_1\phi + (1 - \omega_1)\phi'$. With small enough $\omega_1$, the Q-value of the state-action pairs induced by $\pi_{\phi'}$ (or its slight variant) can be sufficiently evaluated, before being used to update the policy. Next, we detail the loss functions to learn Q-value and policy with neural network approximators.

Denote the parameterized Q-function by $Q_\theta$ with the parameter $\theta$. It is trained by minimizing the Bellman residual of both the AMG and the empirical MDP:

$$L_Q(\theta) = \mathbb{E}_{(s,a,\mathcal{T})\sim\mathcal{D}'}\left[\left(Q_\theta(s,a) - \widehat{Q}_{\text{AMG}}(s,a)\right)^2\right] + \mathbb{E}_{(s,a,s')\sim\mathcal{D}}\left[\left(Q_\theta(s,a) - \widehat{Q}_{\text{MDP}}(s,a)\right)^2\right], \quad (11)$$

with

$$\widehat{Q}_{\text{AMG}}(s,a) = r(s,a) + \gamma\lfloor\min\rfloor_{\tau\in\mathcal{T}}^k \mathbb{E}_\tau\left[\alpha\log\mathbb{E}_{\pi_{\phi'}}\exp\left(\frac{1}{\alpha}Q_{\theta'}(s',a')\right)\right], \quad (12)$$

$$\widehat{Q}_{\text{MDP}}(s,a) = r(s,a) + \gamma\cdot\alpha\log\mathbb{E}_{\pi_{\phi'}}\exp\left(\frac{1}{\alpha}Q_{\theta'}(s',a')\right), \quad (13)$$

where $Q_{\theta'}$ represent the target Q-value softly updated for stability [40], i.e., $\theta' \leftarrow \omega_2\theta + (1 - \omega_2)\theta'$, and $\mathcal{D}'$ is the on-policy data buffer for the AMG. Since the game transition is known, the game can be executed with multiple counterparts in parallel, and the buffer only collects the latest sample for each of them. To promote direct learning from $\mathcal{D}$, we also include the Bellman residual of the empirical MDP in (11).

As with policy update, Theorem 4 states that the optimal policy for $\bar{J}(\pi;\mu)$ is propotional to $\mu(a|s)\exp\left(\frac{1}{\alpha}\bar{Q}_{N,k}^*(s,a)\right)$. Then, we update $\pi_\phi$ by supervisedly learning this policy, with $\mu$ and $\bar{Q}_{N,k}^*$ replaced by the smooth-evolving reference policy and the learned Q-value:

$$L_P(\phi) = \mathbb{E}_{s\sim\mathcal{D}\cup\mathcal{D}'}\left[D_{\text{KL}}\left(\frac{\pi_{\phi'}(\cdot|s)\exp\left(\frac{1}{\alpha}Q_\theta(s,\cdot)\right)}{\mathbb{E}_{\pi_{\phi'}}\left[\exp\left(\frac{1}{\alpha}Q_\theta(s,a)\right)\right]} \,\middle\|\, \pi_\phi(\cdot|s)\right)\right]$$

$$= A\cdot\mathbb{E}_{\substack{s\sim\mathcal{D}\cup\mathcal{D}'\\a\sim\pi_{\phi'}}}\left[\exp\left(\frac{1}{\alpha}Q_\theta(s,a)\right)\log\pi_\phi(a|s)\right] + B, \quad (14)$$

where $A$ and $B$ are constant terms. In general, (14) can be replaced by any tractable function that measures the similarity of distributions. For example, when $\pi_\phi$ is Gaussian, we can apply the recent proposed $\beta$-NLL [41], in which each data point's contribution to the negative log-likelihood loss is weighted by the $\beta$-exponentiated variance to improve learning heteroscedastic behavior.

To summarize, the algorithm alternates between collecting on-policy data samples in AMG and updating the function approximators. In detail, the latter procedure includes updating the Q-value with (11), updating the optimized policy with (14), and updating the target Q-value as well as the reference policy with the moving-average rule. The complete algorithm is listed in Appendix D.

## 5 Experiments

Through the experiments, we aim to answer the following questions: 1) How does the proposed approach compared to the previous SoTA offline RL algorithms on standard benchmark? 2) How does the learning process in the AMG connect with the performance change in the original MDP? 3) Section 3 presents the monotonicity of $J(\pi)$ over $N$ and $k$ for any specified $\pi$, and it is easy to verify that this statement also holds when considering optimal policy for each setting of $(N, k)$. However,

with neural network approximator, our proposed offline RL algorithm approximately solves the AMG. Then, how is the monotonicity satisfied in this case?

We consider the Gym domains in the D4RL benchmark [42] to answer these questions. As PMDB relies on the initial dynamics belief, inserting additional knowledge into the initial dynamics belief will result in unfair comparison. To avoid that, we consider an uniform distribution over dynamics model ensembles as the initial belief. The dynamics model ensembles are trained in supervised manner with the offline dataset. This is similar to previous model-based works [6, 21], where the dynamics model ensembles are considered for dynamics uncertainty quantification. Since hyperparameter tuning for offline RL algorithms requires extra online test for each task, we purposely keep the same hyperparameters for all tasks, except when answering the last question. Especially, the hyperparameters in sampling procedure are $N = 10$ and $k = 2$. The more detailed setup for experiments and hyperparameters can be found in Appendix G. The code is available online[3]

## 5.1 Performance Comparison

We compare the proposed offline RL algorithm with the baselines including: BEAR [5] and BRAC [7], the model-free approaches based on policy constraint, CQL [9], the model-free approach by penalizing Q-value, EDAC [13], the previous SoTA on the D4RL benchmark, MOReL [6], the model-based approach which terminates the trajectory if the dynamics uncertainty exceeds a certain degree, and BC, the behavior cloning method. These approaches are evaluated on a total of eighteen domains involving three environments (hopper, walker2d, halfcheetah) and six dataset types (random, medium, expert, medium-expert, medium-replay, full-replay) per environment. We use the v2 version of each dataset.

The results are summarized in Table 1. Our approach PMDB obviously improves over the previous SoTA on 9 tasks and performs competitively in the rest. Although EDAC achieves better performance in walker2d with several dataset types, its hyperparameters are tuned individually for each task. The later experiments on the impact of hyperparameters indicate that larger $k$ or smaller $N$ could generate better results for walker2d and halfcheetah. We also find that PMDB significantly outperforms MOReL, another model-based approach. It is encouraging that our model-based approach achieves competitive or better performance compared with the SoTA model-free approach, as model-based approach naturally has better support for multi-task learning and transfer learning, where the offline data from relevant tasks can be further leveraged.

| Task Name | BC | BEAR | BRAC | CQL | MOReL | EDAC | PMDB |
|---|---|---|---|---|---|---|---|
| hopper-random | 3.7±0.6 | 3.6±3.6 | 8.1±0.6 | 5.3±0.6 | **38.1±10.1** | 25.3±10.4 | 32.7±0.1 |
| hopper-medium | 54.1±3.8 | 55.3±3.2 | 77.8±6.1 | 61.9±6.4 | 84.0±17.0 | 101.6±0.6 | **106.8±0.2** |
| hopper-expert | 107.7±9.7 | 39.4±20.5 | 78.1±52.6 | 106.5±9.1 | 80.4±34.9 | 110.1±0.1 | **111.7±0.3** |
| hopper-medium-expert | 53.9±4.7 | 66.2±8.5 | 81.3±8.0 | 96.9±15.1 | 105.6±8.2 | 110.7±0.1 | **111.8±0.6** |
| hopper-medium-replay | 16.6±4.8 | 57.7±16.5 | 62.7±30.4 | 86.3±7.3 | 81.8±17.0 | 101.0±0.5 | **106.2±0.6** |
| hopper-full-replay | 19.9±12.9 | 54.0±24.0 | 107.4±0.5 | 101.9±0.6 | 94.4±20.5 | 105.4±0.7 | **109.1±0.2** |
| walker2d-random | 1.3±0.1 | 4.3±1.2 | 1.3±1.4 | 5.4±1.7 | 16.0±7.7 | 16.6±7.0 | **21.8±0.1** |
| walker2d-medium | 70.9±11.0 | 59.8±40.0 | 59.7±39.9 | 79.5±3.2 | 72.8±11.9 | 92.5±0.8 | **94.2±1.1** |
| walker2d-expert | 108.7±0.2 | 110.1±0.6 | 55.2±62.2 | 109.3±0.1 | 62.6±29.9 | 115.1±1.9 | **115.9±1.9** |
| walker2d-medium-expert | 90.1±13.2 | 107.0±2.9 | 9.3±18.9 | 109.1±0.2 | 107.5±5.6 | **114.7±0.9** | 111.9±0.2 |
| walker2d-medium-replay | 20.3±9.8 | 12.2±4.7 | 40.1±47.9 | 76.8±10.0 | 40.8±20.4 | **87.1±2.3** | 79.9±0.2 |
| walker2d-full-replay | 68.8±17.7 | 79.6±15.6 | 96.9±2.2 | 94.2±1.9 | 84.8±13.1 | **99.8±0.7** | 95.4±0.7 |
| halfcheetah-random | 2.2±0.0 | 12.6±1.0 | 24.3±0.7 | 31.3±3.5 | **38.9±1.8** | 28.4±1.0 | 37.8 ± 0.2 |
| halfcheetah-medium | 43.2±0.6 | 42.8±0.1 | 51.9±0.3 | 46.9±0.4 | 60.7±4.4 | 65.9±0.6 | **75.6± 1.3** |
| halfcheetah-expert | 91.8±1.5 | 92.6±0.6 | 39.0±13.8 | 97.3±1.1 | 8.4±11.8 | **106.8±3.4** | 105.7± 1.0 |
| halfcheetah-medium-expert | 44.0±1.6 | 45.7±4.2 | 52.3±0.1 | 95.0±1.4 | 80.4±11.7 | 106.3±1.9 | **108.5±0.5** |
| halfcheetah-medium-replay | 37.6±2.1 | 39.4±0.8 | 48.6±0.4 | 45.3±0.3 | 44.5±5.6 | 61.3±1.9 | **71.7±1.1** |
| halfcheetah-full-replay | 62.9±0.8 | 60.1±3.2 | 78.0±0.7 | 76.9±0.9 | 70.1±5.1 | 84.6±0.9 | **90.0±0.8** |
| Average | 49.9 | 52.4 | 54.0 | 73.7 | 65.1 | 85.2 | **88.2** |

Table 1: Results for D4RL datasets. Each result is the normalized score computed as (score − random policy score) / (expert policy score − random policy score), ± standard deviation. The score of our proposed approach is averaged over 4 random seeds, and the results of the baselines are taken from [13].

## 5.2 Learning in Alternating Markov Game

Figure 2 presents the learning curves in the AMG, as well as the received return when deploying the policy being learned in true MDP. The performance in the AMG closely tracks the true perfor-

---

[3]Code is released at `https://github.com/huawei-noah/HEBO/tree/master/PMDB` and `https://gitee.com/mindspore/models/tree/master/research/rl/pmdb`.

mance from the lower side, implying that it can act as a reasonable surrogate to evaluate/optimize performance for the true MDP. Besides, the performance in the AMG improves nearly monotonously, verifying the effectiveness of the proposed algorithm to approximately solve the game.

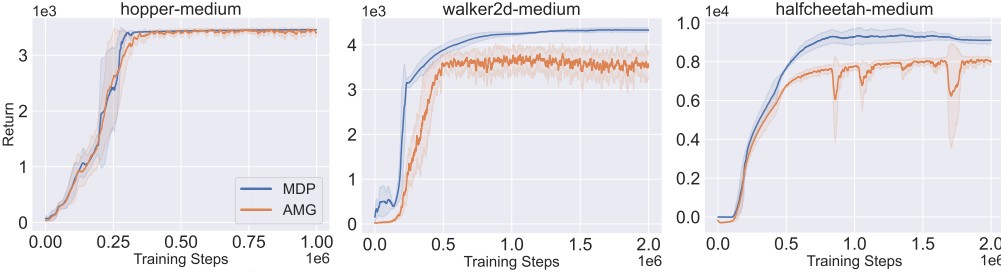

Figure 2: Learning and test curves for medium datasets.

Recall that PMDB does not explicitly quantify dynamics uncertainty to penalize return, Figure 3 checks how the dynamics uncertainty and the Q-value of visited state-action pairs change during the learning process. The uncertainty is measured by the logarithm of standard deviation of the predicted means from the $N$ dynamics samples, i.e., $\log\left(\text{std}\left(\mathbb{E}_\tau[s'];\tau\in\mathcal{T}\right)\right)$. The policy being learned is periodically tested in the AMG for ten trials, and we collect the whole ten trajectories of state-action pairs. The solid curves in Figure 3 denote the mean uncertainty and Q-value over the collected pairs, and shaded regions denote the standard deviation. From the results, the dynamics uncertainty first sharply decreases and then keeps a slowly increasing trend. Besides, in the long-term view, the Q-value is correlated with the degree of uncertainty negatively in the first phase and positively in the second phase. This indicates that the policy first moves to the in-distribution region and then tries to get away by resorting to the generalization of dynamics model.

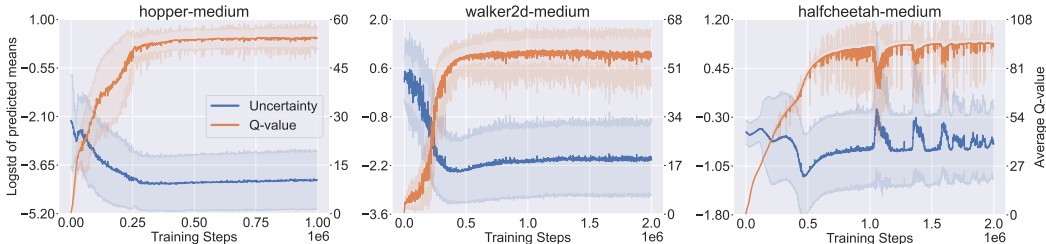

Figure 3: Change on the dynamics uncertainty and Q-value of the encountered state-action pairs during learning process. The dynamics uncertainty of state-action pair $(s, a)$ is measured by $\log\left(\text{std}\left(\mathbb{E}_{\tau(\cdot|s,a)}[s'];\tau\in\mathcal{T}\right)\right)$.

In Figure 3, we also notice that the large dip of Q-value is accompanied with the sudden raise of dynamics uncertainty. We suspect this is due to the optimized policy being far from the dataset. We try to verify by checking the maximal return covered by the offline dataset. It shows that the maximal normalized returns provided by offline datasets are 99.6, 92.0 and 45.0 respectively for hopper, walker2d and halfcheetah, while the proposed approach achieves 106.8, 94.2 and 75.6. The policy optimization is more significant for halfcheetah (where we observe the large dip), indicating the policy should move further from the dataset.

The above finding also explains why the AMG performance in Figure 2 runs into large dip only for halfcheetah: with the larger dynamics uncertainty, the secondary player can choose the more pessimistic transition. However, we want to highlight that it is normal behavior of the proposed algorithm and does not mean instability, as we are handling the alternating Markov game, a specialization of zero-sum game. Besides, we can see that even when the AMG performance goes down the MDP performance is still stable.

## 5.3 Practical Impact of Hyperparameters in Sampling Procedure

Table 2 lists the impact of $k$. In each setting, we evaluate the learned policy in both the true MDP and the AMG. The performance in the AMGs improve when increasing $k$. This is consistent with the theoretical result, even that we approximately solve the game. Regarding the performance in true MDPs, we notice that $k = 2$ corresponds to the best performance for hopper, but for the others $k = 3$ is better. This indicates that tuning hyperparameter online can further improve the performance. The impact of $N$ is presented in Appendix H, suggesting the opposite monotonicity.

| $k$ | hopper-medium | | walker2d-medium | | halfcheetah-medium | |
|---|---|---|---|---|---|---|
| | MDP | AMG | MDP | AMG | MDP | AMG |
| 1 | 106.2±0.2 | 91.6±2.2 | 82.6±0.5 | 33.3±2.6 | 70.7±0.8 | 63.1±0.2 |
| 2 | 106.8±0.2 | 105.2±1.6 | 94.2±1.1 | 77.2±3.7 | 75.6±1.3 | 67.3±1.1 |
| 3 | 90.8±17.5 | 106.6±2.1 | 105.1±0.2 | 82.5±0.5 | 77.3±0.5 | 70.1±0.2 |

Table 2: Impact of $k$, with $N = 10$.

## 6 Related Works

Inadequate data coverage is the root of challenge in offline RL. Existing works differ in their methodology to reacting in face of limited system knowledge.

**Model-free offline RL**  The prevalent idea is to find policy within the data manifold through model-free learning. Analogous to online RL, both policy-based and value-based approaches are devised to this end. Policy-based approaches directly constrain the optimized policy close to the behavior policy that collects data, via various measurements such as KL divergence [7], MMD [5] and action deviation [4, 8]. Value-based approaches instead reflect the policy regularization through value function. For example, CQL enforces small Q-value for OOD state-action pairs [9], AlgaeDICE penalizes return with the $f$-divergence between optimized and offline state-action distributions [11], and Fisher-BRC proposes a novel parameterization of the Q-value to encourage the generated policy close to the data [10]. Our proposed approach is more relevant to the value-based scope, and the key difference to existing works is that our Q-value is penalized through an adversarial choice of transition from plausible candidates.

Learning within the data manifold limits the degree to which the policy improves, and recent works attempt to relieve the restriction. Along the model-free line, EDAC [13] and PBRL [14] quantify uncertainty of Q-value via neural network ensemble, and assign penalty to Q-value depending on the uncertainty degree. In this way, the OOD state-action pairs are touchable if they pose low uncertainty on Q-value. However, the uncertainty quantification over OOD region highly relies on how neural network generalizes [15]. As the prior knowledge of Q-function is hard to acquire and insert into the neural network, the generalization is unlikely reliable to facilitate meaningful uncertainty quantification [16].

**Model-based offline RL**  Model-based approach is widely recognized due to its superior data efficiency. However, directly optimizing policy based on an offline learned model is vulnerable to model exploitation [22, 43]. A line of works improve the dynamics learning for seek of robustness [44] or adaptation [45] to distributional shift. In terms of policy learning, several works extend the idea from model-free approaches, and constrain the optimized policy close to the behavior policy when applying the dynamics model for planing [46] or policy optimization [47, 48]. There are also recent works incorporating uncertainty quantification of dynamics model to learn policy beyond data coverage. Especially, MOPO [21] and MOReL [6] perform policy improvement in states that may not directly occur in the static offline dataset, but can be predicted by leveraging the power of generalization. Compared to them, our approach does not explicitly characterize the dynamics uncertainty as reward penalty. There are also relevant works dealing with model ambiguity in light of Bayesian decision theory, which are discussed in Appendix A.

## 7 Discussion

We proposed model-based offline RL with Pessimism-Modulated Dynamics Belief (PMDB), a framework to reliably learn policy from offline dataset, with the ability of leveraging dynamics prior knowledge. Empirically, the proposed approach outperforms the previous SoTA in a wide range of D4RL tasks. Compared to the previous model-based approaches, we characterize the impact of dynamics uncertainty through biased sampling from the dynamics belief, which implicitly induces PMDB. As PMDB is with the form of reweighting an initial dynamics belief, it provides a principled way to insert prior knowledge via the belief to boost policy learning. However, posing a valuable dynamics belief for arbitrary task is challenging, as the expert knowledge is not always available. Besides, an over-aggressive belief may still incur high-risk behavior in reality. Encouragingly, recent works have done active research to learn data-driven prior from relevant tasks. We believe that integrating them as well as developing safe criterion to design/learn dynamics belief would further promote practical deployment of offline RL.

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
