# A   Additional Related Works

**Robust MDP and CVaR Criterion**   Bayesian decision theory provides a principled formalism for decision making under uncertainty. Robust MDP [23, 24], Conditional Value at Risk (CVaR) [49] and our proposed criterion can be deemed as the specializations of Bayesian decision theory, but derived from different principles and with different properties.

Robust MDP is proposed as a surrogate to optimize the percentile performance. Early works mainly focus on the algorithmic design and theoretical analysis in the tabular case [23, 24] or under linear function approximation [50]. Recently, it has also been extended to continuous action spaces and nonlinear cases, by integrating the advances of deep RL [51, 52]. Meanwhile, a variety of works generalize the uncertainty in regard to system disturbance [53] and action disturbance [54, 55]. Although robust MDP produces robust policy, it purely focuses on the percentile performance, and ignoring the other possibilities is reported to be over-conservative [25–27].

CVaR instead considers the average performance of the worst $\delta$-fraction possibilities. Despite involving more information about the stochasticity, CVaR is still solely from the pessimistic view. Recent works propose to improve by maximizing the convex combination of mean performance and CVaR [25], or maximizing mean performance under CVaR constraint [56]. However, they are intractable regarding policy optimization, i.e., proved as an NP-hard problem or relying on heuristic. As comparison, the proposed AMG formulation presents an alternative way to tackle the entire spectrum of plausible transitions while also give more attention on the pessimistic parts. Besides, the policy optimization is with theoretical guarantee.

Apart from offline RL, Bayesian decision theory is also applied in other RL settings. Particularly, Bayesian RL considers that new observations are continually received and utilized to make adaptive decision. The goal of Bayesian RL is to fast **explore** and adapt, while that of offline RL is to sufficiently **exploit** the offline dataset to generate the best-effort policy supported or surrounded by the dataset. Recently, Bayesian robust RL [57] integrates the idea of robust MDP in the setting of Bayesian RL, where the uncertainty set is constructed to produce robust policy, and will be updated upon new observations to alleviate the degree of conservativeness. Besides, CVaR criterion is also considered in Bayesian RL [58].

# B   Theorem Proof

We first present and prove the fundamental inequalities applied to prove the main theorem, and then present the proofs for Sections 3 and 4 respectively. For conciseness, the subscripts $N, k$ are omitted in Q-value and Bellman backup operator when clear from the context.

## B.1   Preliminaries

**Lemma 1.** *Let $\lfloor \min \rfloor_i^k x_i$ denote the kth minimum in $\{x_i\}$, then*

$$\min_i (x_i - y_i) \leq \lfloor \min \rfloor_i^k x_i - \lfloor \min \rfloor_i^k y_i \leq \max_i (x_i - y_i), \quad \forall k = 1, 2, \cdots, N,$$

*where $N$ is the size of both $\{x_i\}$ and $\{y_i\}$.*

*Proof of Lemma 1.* Denote $i^* = \arg\lfloor \min \rfloor_i^k x_i$ and $j^* = \arg\lfloor \min \rfloor_i^k y_i$. Next, we prove the first inequality. The proof is done by dividing into two cases.

$\boxed{\text{Case 1: } y_{i^*} \geq y_{j^*}}$

It is easy to check

$$\lfloor \min \rfloor_i^k x_i - \lfloor \min \rfloor_i^k y_i = x_{i^*} - y_{j^*} \geq x_{i^*} - y_{i^*} \geq \min_i (x_i - y_i).$$

$\boxed{\text{Case 2: } y_{i^*} < y_{j^*}}$

We prove by contradiction. Let $\mathcal{S}_x = \left\{ \arg\lfloor \min \rfloor_i^l x_i \mid l = 1, 2, \cdots, k-1 \right\}$. Assume

$$y_s < y_{j^*}, \quad \forall s \in \mathcal{S}_x.$$

Since $y_{j^*}$ is the $k$th minimum, the above assumption implies $\mathcal{S}_x \subseteq \mathcal{S}_y$. Meanwhile, according to the condition of case 2, $i^* \in \mathcal{S}_y$. Put these together, we have $\{i^*\} \cup \mathcal{S}_x \subseteq \mathcal{S}_y$. According to the definition of $i^*$, we know $i^* \notin \mathcal{S}_x$. This concludes that $\mathcal{S}_y$ has at least $k$ elements, contradicting with its definition.

Thus,

$$\exists \bar{s} \in \mathcal{S}_x : y_{\bar{s}} \geq y_{j^*}.$$

By applying the above inequality and $x_{\bar{s}} \leq x_{i^*}$, we have

$$\lfloor \min \rfloor_i^k x_i - \lfloor \min \rfloor_i^k y_i = x_{i^*} - y_{j^*} \geq x_{\bar{s}} - y_{\bar{s}} \geq \min_i (x_i - y_i).$$

In summary, we have $\min_i (x_i - y_i) \leq \lfloor \min \rfloor_i^k x_i - \lfloor \min \rfloor_i^k y_i$ for both cases.

The second inequality can be proved by resorting to the first one. By respectively replacing $x_i$ and $y_i$ with $-x_i$ and $-y_i$ in first inequality, we obtain

$$\min_i (-x_i + y_i) \leq \lfloor \min \rfloor_i^k (-x_i) - \lfloor \min \rfloor_i^k (-y_i),$$

which can be rewritten as

$$\max_i (x_i - y_i) \geq - \left( \lfloor \min \rfloor_i^k (-x_i) - \lfloor \min \rfloor_i^k (-y_i) \right)$$
$$= \lfloor \min \rfloor_i^{N-k} x_i - \lfloor \min \rfloor_i^{N-k} y_i,$$

where the last equation is due to $\lfloor \min \rfloor_i^k (-x_i) = -\lfloor \min \rfloor_i^{N-k} x_i$. As the above inequalities holds for any $k \in \{1, 2, \cdots, N\}$, we can replace $N - k$ by $k$, and this is right the second inequality in Lemma 1.

$\square$

**Corollary 1.**

$$\left| \lfloor \min \rfloor_i^k x_i - \lfloor \min \rfloor_i^k y_i \right| \leq \max_i |x_i - y_i|, \quad \forall k = 1, 2, \cdots, N.$$

*Proof of Corollary 1.* The inequality can be attained through simple derivation based on Lemma 1, i.e.,

$$\lfloor \min \rfloor_i^k x_i - \lfloor \min \rfloor_i^k y_i \geq \min_i (x_i - y_i) \geq \min_i (-|x_i - y_i|) = -\max_i |x_i - y_i|$$

and

$$\lfloor \min \rfloor_i^k x_i - \lfloor \min \rfloor_i^k y_i \leq \max_i (x_i - y_i) \leq \max_i |x_i - y_i|.$$

Put them together, we obtain

$$\left| \lfloor \min \rfloor_i^k x_i - \lfloor \min \rfloor_i^k y_i \right| \leq \max_i |x_i - y_i|.$$

$\square$

## B.2 Proofs for Section 3

*Proof of Theorem 1.* Let $Q_1$ and $Q_2$ be two arbitrary Q function, then

$$\|\mathcal{B}^\pi Q_1 - \mathcal{B}^\pi Q_2\|_\infty$$

$$= \gamma \max_{s,a} \left| \mathbb{E}_{\mathbb{P}_T^N}\left[ \lfloor\min\rfloor_{\tau\in\mathcal{T}}^k \mathbb{E}_{\tau,\pi}\left[ Q_1(s',a') \right] \right] - \mathbb{E}_{\mathbb{P}_T^N}\left[ \lfloor\min\rfloor_{\tau\in\mathcal{T}}^k \mathbb{E}_{\tau,\pi}\left[ Q_2(s',a') \right] \right] \right|$$

$$= \gamma \max_{s,a} \left| \mathbb{E}_{\mathbb{P}_T^N}\left[ \lfloor\min\rfloor_{\tau\in\mathcal{T}}^k \mathbb{E}_{\tau,\pi}\left[ Q_1(s',a') \right] - \lfloor\min\rfloor_{\tau\in\mathcal{T}}^k \mathbb{E}_{\tau,\pi}\left[ Q_2(s',a') \right] \right] \right|$$

$$\leq \gamma \max_{s,a} \left( \mathbb{E}_{\mathbb{P}_T^N}\left| \lfloor\min\rfloor_{\tau\in\mathcal{T}}^k \mathbb{E}_{\tau,\pi}\left[ Q_1(s',a') \right] - \lfloor\min\rfloor_{\tau\in\mathcal{T}}^k \mathbb{E}_{\tau,\pi}\left[ Q_2(s',a') \right] \right| \right)$$

$$\leq \gamma \max_{s,a} \left( \mathbb{E}_{\mathbb{P}_T^N}\left[ \max_{\tau\in\mathcal{T}} \left| \mathbb{E}_{\tau,\pi}\left[ Q_1(s',a') - Q_2(s',a') \right] \right| \right] \right)$$

$$\leq \gamma \max_{s,a} \left( \mathbb{E}_{\mathbb{P}_T^N}\|Q_1 - Q_2\|_\infty \right)$$

$$= \gamma\|Q_1 - Q_2\|_\infty,$$

where the second inequality is due to Corollary 1. Thus, the pessimistic Bellman update operator $\mathcal{B}^\pi$ is a contraction mapping.

After convergence, it is easy to check $J(\pi) = \mathbb{E}_{\rho_0,\pi}\left[ Q^\pi(s_0, a_0) \right]$ by recursively unfolding Q-function. $\qquad\square$

*Proof of Theorem 2.* For conciseness, in this proof we drop the superscript $sa$ in $\tau^{sa}$, $\mathbb{P}_T^{sa}$ and $\widetilde{\mathbb{P}}_T^{sa}$.

The proof is based on the definition and the probability density function of order statistic [59]. For any random variables $X_1, X_2, \cdots, X_N$, their $k$th order statistic is defined as $\lfloor\min\rfloor_{n\in\{1,\cdots,N\}}^k X_n$, which is another random variable. Particularly, when $X_1, X_2, \cdots, X_N$ are independent and identically distributed following a probability density function $\mathbb{P}(x)$, the order statistic is with the probability density function

$$\mathbb{P}_{N,k}(x) = \underbrace{\frac{N!}{(k-1)!(N-k)!}}_{C} \mathbb{P}(x)\left[ F(x) \right]^{k-1}\left[ 1 - F(x) \right]^{N-k},$$

where $F(x)$ is the cumulative distribution corresponding to $\mathbb{P}(x)$.

Let $g(\tau) = \mathbb{E}_{\tau,\pi}\left[ Q^\pi(s',a') \right]$ for short. As $\tau$ is random following the belief distribution, $g$ as the functional of $\tau$ is also a random variable. Its sample can be drawn by

$$g = g(\tau), \quad \tau \sim \mathbb{P}_T(\tau).$$

As the elements in $\mathcal{T}$ are independent and identically distributed samples from $\mathbb{P}_T(\tau)$, the elements in $\mathcal{G} = \{g(\tau) \mid \tau \in \mathcal{T}\}$ are also independent and identically distributed. Thus, $\lfloor\min\rfloor_{g\in\mathcal{G}}^k g$ is their $k$th order statistic, and we have

$$\mathbb{E}_{\mathbb{P}_T^N}\left[ \lfloor\min\rfloor_{\tau\in\mathcal{T}}^k g(\tau) \right] = \mathbb{E}_{\mathbb{P}_T^N}\left[ \lfloor\min\rfloor_{g\in\mathcal{G}}^k g \right] = \int_{-\infty}^{\infty} \mathbb{P}_{N,k}(g)g\,dg$$

$$= C\int_{-\infty}^{\infty} \mathbb{P}(g)\left[ F(g) \right]^{k-1}\left[ 1 - F(g) \right]^{N-k} g\,dg,$$

$$= C\int_{-\infty}^{\infty} \left[ \int_{\tau:g(\tau)=g} \mathbb{P}_T(\tau)d\nu(\tau) \right] \left[ F(g) \right]^{k-1}\left[ 1 - F(g) \right]^{N-k} g\,dg,$$

$$= C\int_{-\infty}^{\infty} \int_{\tau:g(\tau)=g} \mathbb{P}_T(\tau)\left[ F(g) \right]^{k-1}\left[ 1 - F(g) \right]^{N-k} g\,d\nu(\tau)dg,$$

$$\overset{(*)}{=} C \int_\tau \int_{g=g(\tau)} \mathbb{P}_T(\tau) \big[ F(g) \big]^{k-1} \big[ 1 - F(g) \big]^{N-k} g \, dg \, d\nu(\tau),$$

$$= \int_\tau \underbrace{C \, \mathbb{P}_T(\tau) \Big[ F\big(g(\tau)\big) \Big]^{k-1} \Big[ 1 - F\big(g(\tau)\big) \Big]^{N-k}}_{\widetilde{\mathbb{P}}_T(\tau)} g(\tau) d\nu(\tau)$$

$$= \mathbb{E}_{\widetilde{\mathbb{P}}_T} [g(\tau)],$$

where $\nu(\tau)$ is the reference measure based on which the belief distribution $P_T^N$ is defined, and the equation $(*)$ is obtained by exchanging the orders of integration. The above equation can rewritten as

$$\mathbb{E}_{\mathbb{P}_T^N} \Big[ \lfloor \min \rfloor_{\tau \in \mathcal{T}}^k \mathbb{E}_{\tau,\pi} \left[ Q(s', a') \right] \Big] = \mathbb{E}_{\widetilde{\mathbb{P}}_T} \mathbb{E}_{\tau,\pi} \left[ Q(s', a') \right].$$

Taking this into consideration, the pessimistic Bellman backup operator in (6) is exactly the vanilla Bellman backup operator for the MDP with transition probability $\widetilde{T}(s'|s, a) = \mathbb{E}_{\widetilde{\mathbb{P}}_T} \left[ \tau(s') \right]$. Then, evaluating/optimizing policy in the AMG is equivalent to evaluating/optimizing in this MDP.

To prove the property of $w$, we treat it as a composite function with form of $w\big(F(x)\big)$. Then, the derivative of $w$ over $F$ is

$$\frac{\delta w}{\delta F} = F^{k-2} (1 - F)^{N-k-1} \left[ (k-1) - (N-1)F \right]. \tag{15}$$

It is easy to check that $\frac{\delta w}{\delta F} \geq 0$ for $F \leq \frac{k-1}{N-1}$ and $\frac{\delta w}{\delta F} \leq 0$ for $F \geq \frac{k-1}{N-1}$. Thus, $w(F)$ reaches the maximum at $F = \frac{k-1}{N-1}$. Besides, as $F(\cdot)$ is the PDF of $x$, it monotonically increases with $x$. Put the monotonicity of $w$ and $F$ together, we know $w(F(x))$ first increases, then decreases with $x$ and achieves the maximimum at $x^* = F^{-1}\left( \frac{k-1}{N-1} \right)$. $\qquad\square$

**Lemma 2** (Monotonicity of Pessimistic Bellman Backup Operator). *Assume that $Q_1 \geq Q_2$ holds element-wisely, then $\mathcal{B}^\pi Q_1 \geq \mathcal{B}^\pi Q_2$ element-wisely.*

*Proof of Lemma 2.*

$$\mathcal{B}^\pi Q_1(s, a) - \mathcal{B}^\pi Q_2(s, a)$$

$$= \gamma \mathbb{E}_{\mathbb{P}_T^N} \left[ \lfloor \min \rfloor_{\tau \in \mathcal{T}}^k \mathbb{E}_{\tau,\pi} \left[ Q_1(s', a') \right] - \lfloor \min \rfloor_{\tau \in \mathcal{T}}^k \mathbb{E}_{\tau,\pi} \left[ Q_2(s', a') \right] \right]$$

$$\geq \gamma \mathbb{E}_{\mathbb{P}_T^N} \left[ \min_{\tau \in \mathcal{T}} \mathbb{E}_{\tau,\pi} \left[ Q_1(s', a') - Q_2(s', a') \right] \right]$$

$$\geq 0, \qquad \forall s, a,$$

where the first inequality is due to Lemma 1. $\qquad\square$

*Proof of Theorem 3.* It is sufficient to prove $Q_{N,k+1}^\pi \geq Q_{N,k}^\pi$, $Q_{N+1,k}^\pi \leq Q_{N,k}^\pi$ and $Q_{N+1,N+1}^\pi \geq Q_{N,N}^\pi$ element-wisely. The idea is to first show $\mathcal{B}_{N,k+1}^\pi Q_{N,k}^\pi \geq Q_{N,k}^\pi$, $\mathcal{B}_{N+1,k}^\pi Q_{N,k}^\pi \leq Q_{N,k}^\pi$ and $\mathcal{B}_{N+1,N+1}^\pi Q_{N,N}^\pi \geq Q_{N,N}^\pi$. Then, the proof can be finished by recursively applying Lemma 2, for example:

$$Q_{N,k+1}^\pi = \lim_{n \to \infty} \big( \mathcal{B}_{N,k+1}^\pi \big)^n Q_{N,k}^\pi \geq \cdots \geq \mathcal{B}_{N,k+1}^\pi Q_{N,k}^\pi \geq Q_{N,k}^\pi.$$

Next, we prove the three inequalities in sequence.

$\boxed{\mathcal{B}_{N,k+1}^{\pi}Q_{N,k}^{\pi} \geq Q_{N,k}^{\pi}}$

$$\mathcal{B}_{N,k+1}^{\pi}Q_{N,k}^{\pi}(s,a)$$

$$= r(s,a) + \gamma\mathbb{E}_{\mathbb{P}_T^n}\left[\lfloor\min\rfloor_{\tau\in\mathcal{T}}^{k+1}\mathbb{E}_{\tau,\pi}\left[Q_{N,k}^{\pi}(s',a')\right]\right]$$

$$\geq r(s,a) + \gamma\mathbb{E}_{\mathbb{P}_T^n}\left[\lfloor\min\rfloor_{\tau\in\mathcal{T}}^{k}\mathbb{E}_{\tau,\pi}\left[Q_{N,k}^{\pi}(s',a')\right]\right]$$

$$= \mathcal{B}_{N,k}^{\pi}Q_{N,k}^{\pi}(s,a)$$

$$= Q_{N,k}^{\pi}(s,a), \qquad \forall s,a,$$

$\boxed{\mathcal{B}_{N+1,k}^{\pi}Q_{N,k}^{\pi} \leq Q_{N,k}^{\pi}}$

$$\mathcal{B}_{N+1,k}^{\pi}Q_{N,k}^{\pi}(s,a)$$

$$= r(s,a) + \gamma\mathbb{E}_{\mathcal{T}\sim\mathbb{P}_T^{N+1}}\left[\lfloor\min\rfloor_{\tau\in\mathcal{T}}^{k}\mathbb{E}_{\tau,\pi}\left[Q_{N,k}^{\pi}(s',a')\right]\right]$$

$$= r(s,a) + \gamma\mathbb{E}_{\mathcal{T}'\sim\mathbb{P}_T^{N}}\left[\mathbb{E}_{\tau'\sim\mathbb{P}_T}\left[\lfloor\min\rfloor_{\tau\in\mathcal{T}'\cup\{\tau'\}}^{k}\mathbb{E}_{\tau,\pi}\left[Q_{N,k}^{\pi}(s',a')\right]\right]\right]$$

$$\leq r(s,a) + \gamma\mathbb{E}_{\mathcal{T}'\sim\mathbb{P}_T^{N}}\left[\lfloor\min\rfloor_{\tau\in\mathcal{T}'}^{k}\mathbb{E}_{\tau,\pi}\left[Q_{N,k}^{\pi}(s',a')\right]\right]$$

$$= \mathcal{B}_{N,k}^{\pi}Q_{N,k}^{\pi}(s,a)$$

$$= Q_{N,k}^{\pi}(s,a), \qquad \forall s,a,$$

where we divide the $(N+1)$-size set $\mathcal{T}$ into $\mathcal{T}'$ and $\{\tau'\}$, $\mathcal{T}'$ contains the first $N$ elements and $\tau'$ is the last element. The second equality is due to the independence among the set elements.

$\boxed{\mathcal{B}_{N+1,N+1}^{\pi}Q_{N,N}^{\pi} \geq Q_{N,N}^{\pi}}$

$$\mathcal{B}_{N+1,N+1}^{\pi}Q_{N,N}^{\pi}(s,a)$$

$$= r(s,a) + \gamma\mathbb{E}_{\mathcal{T}\sim\mathbb{P}_T^{N+1}}\left[\max_{\tau\in\mathcal{T}}\mathbb{E}_{\tau,\pi}\left[Q_{N,k}^{\pi}(s',a')\right]\right]$$

$$= r(s,a) + \gamma\mathbb{E}_{\mathcal{T}'\sim\mathbb{P}_T^{N}}\left[\mathbb{E}_{\tau'\sim\mathbb{P}_T}\left[\max_{\tau\in\mathcal{T}'\cup\{\tau'\}}\mathbb{E}_{\tau,\pi}\left[Q_{N,k}^{\pi}(s',a')\right]\right]\right]$$

$$\geq r(s,a) + \gamma\mathbb{E}_{\mathcal{T}'\sim\mathbb{P}_T^{N}}\left[\max_{\tau\in\mathcal{T}'}\mathbb{E}_{\tau,\pi}\left[Q_{N,k}^{\pi}(s',a')\right]\right]$$

$$= \mathcal{B}_{N,k}^{\pi}Q_{N,k}^{\pi}(s,a)$$

$$= Q_{N,k}^{\pi}(s,a), \qquad \forall s,a,$$

$\square$

## B.3 Proofs for Section 4

Analogous to the policy evaluation for non-regularized case, we define the KL-regularized Bellman update operator for a given policy $\pi$ by

$$\bar{\mathcal{B}}_{N,k}^{\pi}Q(s,a) = r(s,a) + \gamma\mathbb{E}_{\mathbb{P}_T^N}\left[\lfloor\min\rfloor_{\tau\in\mathcal{T}}^{k}\mathbb{E}_{\tau,\pi}\left[Q(s',a') - \alpha D_{\mathrm{KL}}\big(\pi(\cdot|s) \,||\, \mu(\cdot|s)\big)\right]\right]. \quad (16)$$

It is easy to check all proofs in last subsection adapt well for the KL-regularized case. We state the corresponding theorems and lemma as below, and apply them to prove the theorems in Section 4.

**Theorem 6** (Policy Evaluation for KL-Regularized AMG)**.** *The regularized $(N,k)$-pessimistic Bellman backup operator $\bar{\mathcal{B}}_{N,k}^{\pi}$ is a contraction mapping. By starting from any function $Q : \mathcal{S} \times \mathcal{A} \to \mathbb{R}$ and repeatedly applying $\bar{\mathcal{B}}_{N,k}^{\pi}$, the sequence converges to $\bar{Q}_{N,k}^{\pi}$, with which we have*

$$\bar{J}(\pi;\mu) = \mathbb{E}_{\rho_0,\pi}\left[\bar{Q}_{N,k}^{\pi}(s_0,a_0) - \alpha D_{\mathrm{KL}}\big(\pi(\cdot|s_0) \,||\, \mu(\cdot|s_0)\big)\right].$$

**Theorem 7** (Equivalent KL-Regularized MDP with Pessimism-Modulated Dynamics Belief). *The KL-regularized alternating Markov game in (9) is equivalent to the KL-regularized MDP with tuple $(\mathcal{S}, \mathcal{A}, \widetilde{T}, r, \rho_0, \gamma)$, where the transition probability $\widetilde{T}(s'|s, a) = \mathbb{E}_{\widetilde{\mathbb{P}}_T^{sa}}[\tau^{sa}(s')]$ is defined with the reweighted belief distribution $\widetilde{\mathbb{P}}_T^{sa}$:*

$$\widetilde{\mathbb{P}}_T^{sa}(\tau^{sa}) \propto w\Big(\mathbb{E}_{\tau^{sa}, \pi}\big[\bar{Q}_{N,k}^\pi(s', a')\big]; k, N\Big)\mathbb{P}_T^{sa}(\tau^{sa}), \tag{17}$$

$$w(x; k, N) = \big[F(x)\big]^{k-1}\big[1 - F(x)\big]^{N-k}, \tag{18}$$

*and $F(\cdot)$ is cumulative density function. Furthermore, the value of $w(x; k, N)$ first increases and then decreases with $x$, and its maximum is obtained at the $\frac{k-1}{N-1}$ quantile, i.e., $x^* = F^{-1}\left(\frac{k-1}{N-1}\right)$. Similar to the non-regularized case, the reweighting factor $w$ reshapes the initial belief distribution towards being pessimistic in terms of $\mathbb{E}_{\tau, \pi}\big[\bar{Q}_{N,k}^\pi(s', a')\big]$.*

**Lemma 3** (Monotonicity of Regularized Pessimistic Bellman Backup Operator). *Assume that $Q_1 \geq Q_2$ holds element-wisely, then $\bar{\mathcal{B}}_{N,k}^\pi Q_1 \geq \bar{\mathcal{B}}_{N,k}^\pi Q_2$ element-wisely.*

**Theorem 8** (Monotonicity in Regularized Alternating Markov Game). *The converged Q-function $\bar{Q}_{N,k}^\pi$ are with the following properties:*

- *Given any $k$, the Q-function $\bar{Q}_{N,k}^\pi$ element-wisely decreases with $N \in \{k, k+1, \cdots\}$.*
- *Given any $N$, the Q-function $\bar{Q}_{N,k}^\pi$ element-wisely increases with $k \in \{1, 2, \cdots, N\}$.*
- *The Q-function $\bar{Q}_{N,N}^\pi$ element-wisely increases with $N$.*

*Proof of Theorem 4.* The proof of contraction mapping basically follows the same steps in proof of Theorem 1 Let $Q_1$ and $Q_2$ be two arbitrary Q function.

$$\big\|\bar{\mathcal{B}}^* Q_1 - \bar{\mathcal{B}}^* Q_2\big\|_\infty$$

$$= \gamma \max_{s,a} \left| \mathbb{E}_{\mathbb{P}_T^N}\left[ \lfloor \min \rfloor_{\tau \in \mathcal{T}}^k \mathbb{E}_\tau \left[ \alpha \log \mathbb{E}_\mu \exp\left(\frac{1}{\alpha} Q_1(s', a')\right) \right] \right. \right.$$

$$\left. \left. - \lfloor \min \rfloor_{\tau \in \mathcal{T}}^k \mathbb{E}_\tau \left[ \alpha \log \mathbb{E}_\mu \exp\left(\frac{1}{\alpha} Q_2(s', a')\right) \right] \right] \right|$$

$$\leq \gamma \max_{s,a} \left( \mathbb{E}_{\mathbb{P}_T^N}\left| \lfloor \min \rfloor_{\tau \in \mathcal{T}}^k \mathbb{E}_\tau \left[ \alpha \log \mathbb{E}_\mu \exp\left(\frac{1}{\alpha} Q_1(s', a')\right) \right] \right. \right.$$

$$\left. \left. - \lfloor \min \rfloor_{\tau \in \mathcal{T}}^k \mathbb{E}_\tau \left[ \alpha \log \mathbb{E}_\mu \exp\left(\frac{1}{\alpha} Q_2(s', a')\right) \right] \right| \right)$$

$$\leq \gamma \max_{s,a} \left( \mathbb{E}_{\mathbb{P}_T^N}\left[ \max_{\tau \in \mathcal{T}} \left| \mathbb{E}_\tau \left[ \alpha \log \mathbb{E}_\mu \exp\left(\frac{1}{\alpha} Q_1(s', a')\right) \right] - \mathbb{E}_\tau \left[ \alpha \log \mathbb{E}_\mu \exp\left(\frac{1}{\alpha} Q_2(s', a')\right) \right] \right| \right] \right)$$

$$= \gamma \max_{s,a} \left( \mathbb{E}_{\mathbb{P}_T^N}\left[ \max_{\tau \in \mathcal{T}} \left| \mathbb{E}_\tau \left[ \alpha \log \mathbb{E}_\mu \exp\left(\frac{1}{\alpha} Q_1(s', a')\right) - \alpha \log \mathbb{E}_\mu \exp\left(\frac{1}{\alpha} Q_2(s', a')\right) \right] \right| \right] \right)$$

$$\leq \gamma \max_{s,a} \left( \mathbb{E}_{\mathbb{P}_T^N} \|Q_1 - Q_2\|_\infty \right)$$

$$= \gamma \|Q_1 - Q_2\|_\infty,$$

where the second inequality is obtained with Corollary 1, and the last inequality is due to $\big\|\alpha \log \mathbb{E}_\mu \exp\big(\frac{1}{\alpha} Q_1(s, a)\big) - \alpha \log \mathbb{E}_\mu \exp\big(\frac{1}{\alpha} Q_2(s, a)\big)\big\|_\infty \leq \|Q_1 - Q_2\|_\infty$. We present its proof by following [34]:

Suppose $\epsilon = \|Q_1 - Q_2\|_\infty$, then

$$\alpha \log \mathbb{E}_\mu \exp\left(\frac{1}{\alpha} Q_1(s, a)\right) \leq \alpha \log \mathbb{E}_\mu \exp\left(\frac{1}{\alpha} Q_2(s, a) + \frac{\epsilon}{\alpha}\right)$$

$$= \alpha \log \mathbb{E}_\mu \exp\left(\frac{1}{\alpha} Q_2(s, a)\right) + \epsilon.$$

Similarly, $\alpha \log \mathbb{E}_\mu \exp\left(\frac{1}{\alpha}Q_1(s,a)\right) \geq \alpha \log \mathbb{E}_\mu \exp\left(\frac{1}{\alpha}Q_2(s,a)\right) - \epsilon$. The desired inequality is proved by putting them together.

Next, we prove $\bar{\pi}^*(a|s) \propto \mu(a|s)\exp\left(\frac{1}{\alpha}\bar{Q}^*(s,a)\right)$ is the optimal policy for $\bar{J}(\pi;\mu)$.

First, for any policy $\pi'$,

$$\bar{\mathcal{B}}^* \bar{Q}^{\pi'}(s,a)$$

$$= r(s,a) + \gamma \mathbb{E}_{\mathbb{P}_T^N}\left[\lfloor\min\rfloor_{\tau\in\mathcal{T}}^k \mathbb{E}_\tau\left[\alpha \log \mathbb{E}_\mu \exp\left(\frac{1}{\alpha}\bar{Q}^{\pi'}(s',a')\right)\right]\right]$$

$$= r(s,a) + \gamma \mathbb{E}_{\mathbb{P}_T^N}\left[\lfloor\min\rfloor_{\tau\in\mathcal{T}}^k \mathbb{E}_\tau\left[\alpha \log \mathbb{E}_\mu \exp\left(\frac{1}{\alpha}\bar{Q}^{\pi'}(s',a')\right)\right.\right.$$
$$\left.\left. - \min_\pi \alpha D_{\mathrm{KL}}\left(\pi(\cdot|s') \,\Big|\Big|\, \frac{\mu(\cdot|s')\exp\frac{1}{\alpha}\bar{Q}^{\pi'}(s',\cdot)}{\mathbb{E}_\mu \exp\left(\frac{1}{\alpha}\bar{Q}^{\pi'}(s',a')\right)}\right)\right]\right]$$

$$= r(s,a) + \gamma \mathbb{E}_{\mathbb{P}_T^N}\left[\lfloor\min\rfloor_{\tau\in\mathcal{T}}^k \mathbb{E}_\tau\left[\max_\pi\left(\mathbb{E}_\pi\left[\bar{Q}^{\pi'}(s',a')\right] - \alpha D_{\mathrm{KL}}\left(\pi(\cdot|s') \,\big|\big|\, \mu(\cdot|s')\right)\right)\right]\right]$$

$$\geq r(s,a) + \gamma \mathbb{E}_{\mathbb{P}_T^N}\left[\lfloor\min\rfloor_{\tau\in\mathcal{T}}^k \mathbb{E}_\tau\left[\mathbb{E}_{\pi'}\left[\bar{Q}^{\pi'}(s',a')\right] - \alpha D_{\mathrm{KL}}\left(\pi'(\cdot|s') \,\big|\big|\, \mu(\cdot|s')\right)\right]\right]$$

$$= \bar{\mathcal{B}}^{\pi'} \bar{Q}^{\pi'}(s,a)$$

$$= \bar{Q}^{\pi'}(s,a), \qquad \forall s,a.$$

By applying Lemma 3 recursively, we obtain

$$\bar{Q}^*(s,a) = \lim_{n\to\infty}\left(\bar{\mathcal{B}}^*\right)^n \bar{Q}^{\pi'}(s,a) \geq \cdots \geq \bar{\mathcal{B}}^* \bar{Q}^{\pi'}(s,a) \geq \bar{Q}^{\pi'}(s,a), \quad \forall s,a. \qquad (19)$$

Besides,

$$\bar{\mathcal{B}}^{\bar{\pi}^*} \bar{Q}^*(s,a)$$

$$= r(s,a) + \gamma \mathbb{E}_{\mathbb{P}_T^N}\left[\lfloor\min\rfloor_{\tau\in\mathcal{T}}^k \mathbb{E}_{\tau,\bar{\pi}^*}\left[\bar{Q}^*(s',a') - \alpha D_{\mathrm{KL}}\left(\bar{\pi}^*(\cdot|s) \,\big|\big|\, \mu(\cdot|s)\right)\right]\right]$$

$$= r(s,a) + \gamma \mathbb{E}_{\mathbb{P}_T^N}\left[\lfloor\min\rfloor_{\tau\in\mathcal{T}}^k \mathbb{E}_\tau\left[\alpha \log \mathbb{E}_\mu \exp\left(\frac{1}{\alpha}\bar{Q}^*(s',a')\right)\right]\right]$$

$$= \bar{Q}^*(s,a), \qquad \forall s,a.$$

By repeatedly applying $\bar{\mathcal{B}}^{\bar{\pi}^*}$ to the above equation, we obtain

$$\bar{Q}^{\bar{\pi}^*}(s,a) = \lim_{n\to\infty}\left(\bar{\mathcal{B}}^{\bar{\pi}^*}\right)^n \bar{Q}^*(s,a) = \cdots = \bar{\mathcal{B}}^{\bar{\pi}^*} \bar{Q}^*(s,a) = \bar{Q}^*(s,a), \quad \forall s,a. \qquad (20)$$

By combining equations (19) and (20), we have

$$\bar{Q}^{\bar{\pi}^*}(s,a) \geq \bar{Q}^{\pi'}(s,a), \quad \forall \pi', \forall s,a. \qquad (21)$$

Finally, by expanding $\bar{J}$ as stated in Theorem 6 and applying (21), the proof is completed

$$\bar{J}(\bar{\pi}^*;\mu) = \mathbb{E}_{\rho_0,\bar{\pi}^*}\left[\bar{Q}^{\bar{\pi}^*}(s_0,a_0) - \alpha D_{\mathrm{KL}}\left(\bar{\pi}^*(\cdot|s_0) \,\big|\big|\, \mu(\cdot|s_0)\right)\right]$$

$$\geq \mathbb{E}_{\rho_0,\bar{\pi}^*}\left[\bar{Q}^{\pi'}(s_0,a_0) - \alpha D_{\mathrm{KL}}\left(\bar{\pi}^*(\cdot|s_0) \,\big|\big|\, \mu(\cdot|s_0)\right)\right]$$

$$\geq \mathbb{E}_{\rho_0,\pi'}\left[\bar{Q}^{\pi'}(s_0,a_0) - \alpha D_{\mathrm{KL}}\left(\pi'(\cdot|s_0) \,\big|\big|\, \mu(\cdot|s_0)\right)\right]$$

$$= \bar{J}(\pi';\mu), \quad \forall \pi'.$$

$\square$

*Proof of Theorem 5.* We first prove $J(\pi_{i+1}) > J(\pi_i)$. As the iteration requires $\bar{J}(\pi_{i+1}; \pi_i) > \bar{J}(\pi_i; \pi_i) = J(\pi_i)$, it is sufficient to prove $J(\pi_{i+1}) \geq \bar{J}(\pi_{i+1}; \pi_i)$. We do that by showing $Q^{\pi_{i+1}} \geq \bar{Q}^{\pi_{i+1}}$ element-wisely.

First,

$$
\begin{aligned}
&\mathcal{B}^{\pi_{i+1}} \bar{Q}^{\pi_{i+1}}(s, a) - \bar{Q}^{\pi_{i+1}}(s, a) \\
&= \mathcal{B}^{\pi_{i+1}} \bar{Q}^{\pi_{i+1}}(s, a) - \bar{\mathcal{B}}^{\pi_{i+1}} \bar{Q}^{\pi_{i+1}}(s, a) \\
&= \gamma \mathbb{E}_{\mathbb{P}_T^N} \left[ \lfloor \min \rfloor_{\tau \in \mathcal{T}}^k \mathbb{E}_{\tau, \pi_{i+1}} \left[ \bar{Q}^{\pi_{i+1}}(s', a') \right] \right] \\
&\quad - \gamma \mathbb{E}_{\mathbb{P}_T^N} \left[ \lfloor \min \rfloor_{\tau \in \mathcal{T}}^k \mathbb{E}_{\tau, \pi_{i+1}} \left[ \bar{Q}^{\pi_{i+1}}(s', a') - \alpha D_{\mathrm{KL}}\big(\pi_{i+1}(\cdot|s') \,||\, \pi_i(\cdot|s')\big) \right] \right] \\
&\geq \gamma \mathbb{E}_{\mathbb{P}_T^N} \left[ \min_{\tau} \mathbb{E}_{\tau} \left[ \alpha D_{\mathrm{KL}}\big(\pi_{i+1}(\cdot|s') \,||\, \pi_i(\cdot|s')\big) \right] \right] \\
&\geq 0, \qquad \forall s, a,
\end{aligned}
\tag{22}
$$

where the first inequality is due to Lemma 1, the second inequality is due to the non-negativity of KL-divergence.

Then, by recursively applying Lemma 2 we obtain

$$
Q^{\pi_{i+1}}(s, a) = \lim_{n \to \infty} (\mathcal{B}^{\pi_{i+1}})^n \bar{Q}^{\pi_{i+1}}(s, a) \geq \cdots \geq \mathcal{B}^{\pi_{i+1}} \bar{Q}^{\pi_{i+1}}(s, a) \geq \bar{Q}^{\pi_{i+1}}(s, a), \quad \forall s, a.
\tag{23}
$$

By substituting into $J(\pi_{i+1})$ and $\bar{J}(\pi_{i+1}; \pi_i)$, we have

$$
\begin{aligned}
J(\pi_{i+1}) &= \mathbb{E}_{\rho_0, \pi_{i+1}} Q^{\pi_{i+1}}(s_0, a_0) \\
&\geq \mathbb{E}_{\rho_0, \pi_{i+1}} \bar{Q}^{\pi_{i+1}}(s_0, a_0) \\
&\geq \mathbb{E}_{\rho_0, \pi_{i+1}} \left[ \bar{Q}^{\pi_{i+1}}(s_0, a_0) - \alpha D_{\mathrm{KL}}(\pi_{i+1}(\cdot|s_0) \,||\, \pi_i(\cdot|s_0)) \right] \\
&= \bar{J}(\pi_{i+1}; \pi_i).
\end{aligned}
\tag{24}
$$

To summarize, the proof is done via $J(\pi_{i+1}) \geq \bar{J}(\pi_{i+1}; \pi_i) > \bar{J}(\pi_i; \pi_i) = J(\pi_i)$.

Next, we consider the special case where $\{\pi_i\}$ are obtained via regularized policy optimization in Theorem 4. For the $(i+1)$th step, $\pi_{i+1}$ is the optimal solution for the sub-problem of maximizing $J(\pi; \pi_i)$. Thus, according to (21), $\bar{Q}^{\pi_{i+1}}(s, a) \geq \bar{Q}^{\pi'}(s, a), \forall \pi', \forall s, a$. For $\pi' = \pi_i$, the KL term in Q-value vanishes and we have $\bar{Q}^{\pi_{i+1}}(s, a) \geq Q^{\pi_i}(s, a)$. By combining it with (23), we obtain

$$
Q^{\pi_{i+1}}(s, a) \geq \bar{Q}^{\pi_{i+1}}(s, a) \geq Q^{\pi_i}(s, a), \quad \forall s, a.
\tag{25}
$$

Then, the boundness of $Q$ indicates the existence of $\lim_{i \to \infty} Q^{\pi_i}(s, a)$ and also $\lim_{i \to \infty} Q^{\pi_i}(s, a) = \lim_{i \to \infty} \bar{Q}^{\pi_i}(s, a), \forall s, a$.

For any $s, a, a'$ satisfying $\lim_{i \to \infty} Q^{\pi_i}(s, a) > \lim_{i \to \infty} Q^{\pi_i}(s, a')$, it satisfies $\lim_{i \to \infty} \bar{Q}^{\pi_i}(s, a) > \lim_{i \to \infty} \bar{Q}^{\pi_i}(s, a')$. Thus,

$$
\exists N, \epsilon > 0 \quad \forall j \geq N : \bar{Q}^{\pi_j}(s, a) - \bar{Q}^{\pi_j}(s, a') \geq \epsilon.
\tag{26}
$$

According to Theorem 4, the updated policy is with form of[4]

$$
\pi_i(a|s) \propto \pi_{i-1}(a|s) \exp\left( \frac{1}{\alpha} \bar{Q}^{\pi_i}(s, a) \right).
$$

Then, the policy ratio can be rewritten and bounded as

$$
\frac{\pi_i(a|s)}{\pi_i(a'|s)} = \frac{\pi_N(a|s)}{\pi_N(a'|s)} \exp\left( \sum_{j=N}^{i} \frac{\bar{Q}^{\pi_j}(s, a) - \bar{Q}^{\pi_j}(s, a')}{\alpha} \right) \geq \frac{\pi_N(a|s)}{\pi_N(a'|s)} \exp\left( \frac{i - N}{\alpha} \epsilon \right), \quad \forall i \geq N.
\tag{27}
$$

With the prerequisite of $\pi_0(a|s) > 0, \forall s, a$ and the form of policy update, we know $\pi_N(a|s) > 0, \forall s, a$, and further $\frac{\pi_N(a|s)}{\pi_N(a'|s)} > 0$. Then, as $i$ approaches infinity in (27), we obtain $\frac{\pi_i(a|s)}{\pi_i(a'|s)} \to \infty$. $\square$

---

[4] Strictly speaking, Theorem 4 shows $\bar{\pi}^*(a|s) \propto \mu(a|s) \exp\left( \frac{1}{\alpha} \bar{Q}_{N,k}^*(s, a) \right)$. Besides, we have shown $\bar{Q}_{N,k}^{\bar{\pi}^*}(s, a) = \bar{Q}_{N,k}^*(s, a)$ in (20). Thus, $\bar{\pi}^*(a|s) \propto \mu(a|s) \exp\left( \frac{1}{\alpha} \bar{Q}_{N,k}^{\bar{\pi}^*}(s, a) \right)$.

# C  Iterative Regularized Policy Optimization as Expectation–Maximization with Structured Variational Posterior

This section recasts the iterative regularized policy optimization as an Expectation-Maximization algorithm for policy optimization, where the Expectation step corresponds to a structured variational inference procedure for dynamics. To simplify the presentation, we consider the $L$-length horizon and let $\gamma = 1$ (thus omitted in the derivation). For infinite horizon $L \to \infty$, the discounted factor $\gamma$ can be readily recovered by modifying the transition dynamics, such that any action produces a transition into an terminal state with probability $1 - \gamma$.

## C.1  Review of RL as Probabilistic Inference

We first review the general framework of casting RL as probabilistic inference [60]. It starts by embedding the MDP into a probabilistic graphical model, as shown in Figure 4. Apart from the basic elements in MDP, an additional binary random variable $\mathcal{O}_t$ is introduced, where $\mathcal{O}_t = 1$ denotes that the action at time step $t$ is optimal, and $\mathcal{O}_t = 0$ denotes the suboptimality. Its distribution is defined as[5]

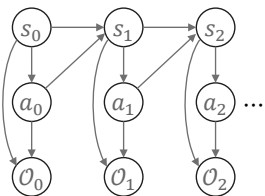

$$p(\mathcal{O}_t = 1 | s_t, a_t) = \exp\left(\frac{r(s_t, a_t)}{\alpha}\right), \qquad (28)$$

Figure 4: Probabilistic graphical model for RL as inference.

where $\alpha$ is the hyperparameter. As we focus on the optimality, in the following we drop $= 1$ and use $\mathcal{O}_t$ to denote $\mathcal{O}_t = 1$ for conciseness. The remaining random variables in the probabilistic graphical model are $s_t$ and $a_t$, whose distributions are defined by the system dynamics $\rho_0(s)$ and $T(s'|s, a)$ as well as a reference policy $\mu(a|s)$. Then, the joint distribution over all random variables for $t \in \{1, 2, \cdots, L\}$ can be written as

$$\mathbb{P}\left(s_{0:L}, a_{0:L}, \mathcal{O}_{0:L}\right) = \rho_0(s_0) \cdot \prod_{t=0}^{L-1} T(s_{t+1}|s_t, a_t) \mu(a_t|s_t) \cdot \mu(a_L|s_L) \exp\left(\sum_{t=0}^{L} \frac{r(s_t, a_t)}{\alpha}\right). \quad (29)$$

Regarding optimal control, a natural question to ask is what the trajectory should be like given the optimality over all time steps. This raises the posterior inference of $\mathbb{P}\left(s_{0:L}, a_{0:L}|\mathcal{O}_{0:L}\right)$. According to d-separation, the exact posterior follows the form of

$$\mathbb{P}\left(s_{0:L}, a_{0:L}|\mathcal{O}_{0:L}\right) = \mathbb{P}(s_0|\mathcal{O}_{0:L}) \cdot \prod_{t=0}^{L-1} \mathbb{P}(s_{t+1}|s_t, a_t, \mathcal{O}_{0:L}) \mathbb{P}(a_t|s_t, \mathcal{O}_{0:L}) \cdot \mathbb{P}(a_L|s_L, \mathcal{O}_{0:L}).$$

$$(30)$$

Notice that the dynamics posterior $\mathbb{P}(s_0|\mathcal{O}_{0:L})$ and $\mathbb{P}(s_{t+1}|s_t, a_t, \mathcal{O}_{0:L})$ depends on $\mathcal{O}_{0:L}$, and in fact their concrete mathematical expressions are inconsistent with those of the system dynamics $\rho_0(s_0)$ and $T(s_{t+1}|s_t, a_t)$ [60]. This essentially poses the assumption that the dynamics itself can be controlled when referring to the optimality, unpractical in general.

Variational inference can be applied to correct this issue. Concretely, define the variational approximation to the exact posterior by

$$\widehat{\mathbb{P}}\left(s_{0:L}, a_{0:L}\right) = \rho_0(s_0) \cdot \prod_{t=0}^{L-1} T(s_{t+1}|s_t, a_t) \pi(a_t|s_t) \cdot \pi(a_L|s_L). \qquad (31)$$

Its difference to (30) is enforcing the dynamics posterior to match the practical one. Under this structure, the variational posterior can be adjusted by optimizing $\pi$ to best approximate the exact

---

[5]Assume the reward function is non-positive such that the probability is not larger than one. If the assumption is unsatisfied, we can subtract the reward function by its maximum, without changing the optimal policy.

posterior. The optimization is executed under measure of KL divergence, i.e.,

$$D_{\mathrm{KL}}\left(\widehat{\mathbb{P}}\left(s_{0:L}, a_{0:L}\right) \,\middle|\middle|\, \mathbb{P}\left(s_{0:L}, a_{0:L}|\mathcal{O}_{0:L}\right)\right) = \int \widehat{\mathbb{P}}\left(s_{0:L}, a_{0:L}\right) \log \frac{\widehat{\mathbb{P}}\left(s_{0:L}, a_{0:L}\right)}{\mathbb{P}\left(s_{0:L}, a_{0:L}|\mathcal{O}_{0:L}\right)} ds_{0:L} da_{0:L}$$

$$= \int \widehat{\mathbb{P}}\left(s_{0:L}, a_{0:L}\right) \log \frac{\widehat{\mathbb{P}}\left(s_{0:L}, a_{0:L}\right)}{\mathbb{P}\left(s_{0:L}, a_{0:L}, \mathcal{O}_{0:L}\right)} ds_{0:L} da_{0:L} + \log \mathbb{P}(\mathcal{O}_{0:L})$$

$$= \mathbb{E}_{\rho_0, T, \pi}\left[\sum_{t=0}^{L}\left(-\frac{r(s_t, a_t)}{\alpha} + \log \frac{\pi(a_t|s_t)}{\mu(a_t|s_t)}\right)\right] + \log \mathbb{P}(\mathcal{O}_{0:L})$$

$$= \frac{1}{\alpha}\mathbb{E}_{\rho_0, T, \pi}\left[\sum_{t=0}^{L}\left(-r(s_t, a_t) + \alpha D_{\mathrm{KL}}\left(\pi(\cdot|s_t) \,\middle|\middle|\, \mu(\cdot|s_t)\right)\right)\right] + \log \mathbb{P}(\mathcal{O}_{0:L}), \tag{32}$$

where the third equation is obtained by substituting (29) and (31). As the second term in (32) is constant, minimizing the above KL divergence is equivalent to maximize the cumulative reward with policy regularizer. Several fascinating online RL methods can be treated as algorithmic instances based on this framework [34, 35].

To summarize, the structured variational posterior with form (31) is vital to ensure the inferred policy meaningful in the actual environment.

## C.2  Pessimism-Modulated Dynamics Belief as Structured Variational Posterior

The probabilistic graphical model is previously devised for online RL. In offline setting, the environment can not be interacted to minimize (32). A straightforward modification to reflect this is to add the transition dynamics as a random variable in the graph, as shown in Figure 5. We assume the transition follows a predefined belief distribution, i.e., $\mathbb{P}_T^{sa}(\tau^{sa})$ introduced in Subsection 3.1. To make its dependence on $(s, a)$ explicit, let $\mathbb{P}_T(\tau^{sa}|s, a)$ redenote $\mathbb{P}_T^{sa}(\tau^{sa})$. For conciseness, we drop the superscript $sa$ in $\tau^{sa}$ in the remainder.

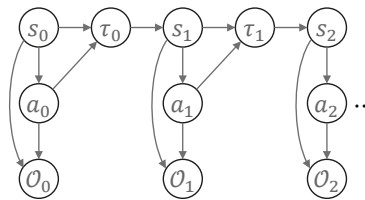

Figure 5: Probabilistic graphical model for offline RL as inference.

The joint distribution over all random variables in Figure 5 for $t \in \{1, 2, \cdots, L\}$ can be written as

$$\mathbb{P}\left(s_{0:L}, a_{0:L}, \tau_{0:L-1}, \mathcal{O}_{0:L}\right) = \rho_0(s_0) \cdot \prod_{t=0}^{L-1} \mathbb{P}_T(\tau_t|s_t, a_t)\tau_t(s_{t+1})\mu(a_t|s_t)$$

$$\cdot \mu(a_L|s_L)\exp\left(\sum_{t=0}^{L}\frac{r(s_t, a_t)}{\alpha}\right). \tag{33}$$

Similar to online setting, we wonder what the trajectory should be like given the optimality over all time steps. By examining the conditional independence in the probabilistic graphical model, the exact posterior follows the form of

$$\mathbb{P}\left(s_{0:L}, a_{0:L}, \tau_{0:L-1}|\mathcal{O}_{0:L}\right) = \mathbb{P}(s_0|\mathcal{O}_{0:L}) \cdot \prod_{t=0}^{L-1} \mathbb{P}(\tau_t|s_t, a_t, \mathcal{O}_{0:L})\mathbb{P}(s_{t+1}|\tau_t, \mathcal{O}_{0:L})\mathbb{P}(a_t|s_t, \mathcal{O}_{0:L})$$

$$\cdot \mathbb{P}(a_L|s_L, \mathcal{O}_{0:L}). \tag{34}$$

Unsurprisingly, $s_{0:T}$ and $\tau_{0:T}$ again depend on $\mathcal{O}_{0:L}$, indicating that the system transition and its belief can be controlled when referring to optimality. In other words, it leads to over-optimistic inference.

To emphasize pessimism, we define a novel structured variational posterior:

$$\widehat{\mathbb{P}}\left(s_{0:L}, a_{0:L}, \tau_{0:L-1}\right) = \rho_0(s_0) \cdot \prod_{t=0}^{L-1} \widetilde{\mathbb{P}}_T(\tau_t|s_t, a_t)\tau_t(s_{t+1})\pi(a_t|s_t) \cdot \pi(a_L|s_L), \tag{35}$$

with $\widetilde{\mathbb{P}}_T$ being the *Pessimism-Modulated Dynamics Belief (PMDB)* constructed via the KL-regularized AMG (see Theorem 7):

$$\widetilde{\mathbb{P}}_T(\tau|s,a) \propto w\Big(\mathbb{E}_{\tau,\pi}\big[\bar{Q}_{N,k}^{\pi}(s',a')\big]; k, N\Big)\mathbb{P}_T(\tau|s,a), \tag{36}$$

$$w(x;k,N) = \big[F(x)\big]^{k-1}\big[1-F(x)\big]^{N-k}, \tag{37}$$

$F(\cdot)$ is cumulative density function and $\bar{Q}_{N,k}^{\pi}$ is the Q-value for the KL-regularized AMG. As discussed, $w$ reshapes the initial belief distribution towards being pessimistic in terms of $\mathbb{E}_{\tau,\pi}\big[\bar{Q}_{N,k}^{\pi}(s',a')\big]$.

It seems that we need to solve the AMG to obtain $\bar{Q}_{N,k}^{\pi}$ and further define $\widetilde{\mathbb{P}}_T$. In fact, $\bar{Q}_{N,k}^{\pi}$ is also the Q-value for the MDP considered in (35). This can be verified by checking Theorem 7: the KL-regularized AMG is equivalent to the MDP with transition $\widetilde{T}(s'|s,a) = \mathbb{E}_{\widetilde{\mathbb{P}}_T}[\tau(s')]$, which can be implemented by sampling first $\tau \sim \widetilde{\mathbb{P}}_T$ and then $s' \sim \tau$, right the procedure in (35).

To best approximate the exact posterior, we optimize the variational posterior by minimizing

$$D_{\mathrm{KL}}\Big(\widehat{\mathbb{P}}\left(s_{0:L}, a_{0:L}, \tau_{0:L-1}\right) \,\Big\|\, \mathbb{P}\left(s_{0:L}, a_{0:L}, \tau_{0:L-1}|\mathcal{O}_{0:L}\right)\Big)$$

$$= \int \widehat{\mathbb{P}}\left(s_{0:L}, a_{0:L}, \tau_{0:L-1}\right) \log \frac{\widehat{\mathbb{P}}\left(s_{0:L}, a_{0:L}, \tau_{0:L-1}\right)}{\mathbb{P}\left(s_{0:L}, a_{0:L}, \tau_{0:L-1}|\mathcal{O}_{0:L}\right)} ds_{0:L} da_{0:L}$$

$$= \int \widehat{\mathbb{P}}\left(s_{0:L}, a_{0:L}, \tau_{0:L-1}\right) \log \frac{\widehat{\mathbb{P}}\left(s_{0:L}, a_{0:L}, \tau_{0:L-1}\right)}{\mathbb{P}\left(s_{0:L}, a_{0:L}, \tau_{0:L-1}, \mathcal{O}_{0:L}\right)} ds_{0:L} da_{0:L} + \log \mathbb{P}(\mathcal{O}_{0:L})$$

$$= \underbrace{\mathbb{E}_{\rho_0, \widetilde{\mathbb{P}}_T, \tau_{0:L-1}, \pi}\left[\sum_{t=0}^{L}\left(-\frac{r(s_t, a_t)}{\alpha} + D_{\mathrm{KL}}\Big(\pi(\cdot|s_t)\,\Big\|\,\mu(\cdot|s_t)\Big)\right)\right]}_{M(\pi;\mu)}$$

$$+ \mathbb{E}_{\rho_0, \widetilde{\mathbb{P}}_T, \tau_{0:L-1}, \pi}\left[\sum_{t=0}^{L-1} \log w\Big(\mathbb{E}_{\tau_t,\pi}\big[\bar{Q}_{N,k}^{\pi}(s_{t+1}, a_{t+1})\big]; k, N\Big)\right] + (L-1) \cdot \log C$$

$$\stackrel{(*)}{=} M(\pi;\mu) + (L-1) \cdot \log C$$

$$+ \sum_{t=0}^{L-1} \mathbb{E}_{\rho_0, \pi, \widehat{\mathbb{P}}_T}\left[\mathbb{E}_{\tau_0, \pi, \widehat{\mathbb{P}}_T} \cdots \left[\mathbb{E}_{\tau_{t-1}, \pi, \widetilde{\mathbb{P}}_T}\left[\log w\Big(\mathbb{E}_{\tau_t,\pi}\big[\bar{Q}_{N,k}^{\pi}(s_{t+1}, a_{t+1})\big]; k, N\Big)\right]\right]\right]$$

$$\approx M(\pi;\mu) + (L-1) \cdot (\log C' + \log C), \tag{38}$$

where the equation $(*)$ is by unfolding the expectation sequentially over each step, $C = \frac{N!}{(k-1)!(N-k)!}$ is the normalization constant in (36), and $C' = \frac{(k-1)^{k-1}(N-k)^{N-k}}{(N-1)^{N-1}}$ is used to approximate $w$. To clarify the approximation, recall Theorem 7 stating that a sample $\tau_t \sim \widetilde{\mathbb{P}}_T$ can be equivalently drawn by finding $\tau_t = \arg\lfloor\min\rfloor_{\tau \in \mathcal{T}_t}^{k}\mathbb{E}_{\tau,\pi}\big[\bar{Q}_{N,k}^{\pi}(s_{t+1}, a_{t+1})\big]$ based on another sampling procedure $\mathcal{T}_t = \{\tau\}^N \sim \mathbb{P}_T^N$. Then, given $\mathcal{T}_t$, we observe that $\mathbb{E}_{\tau_t,\pi}\big[\bar{Q}_{N,k}^{\pi}(s_{t+1}, a_{t+1})\big]$ is the empirical $\frac{k-1}{N-1}$ quantile of the random variable $\mathbb{E}_{\tau,\pi}\big[\bar{Q}_{N,k}^{\pi}(s_{t+1}, a_{t+1})\big]$, i.e., $F\left(\mathbb{E}_{\tau_t,\pi}\big[\bar{Q}_{N,k}^{\pi}(s_{t+1}, a_{t+1})\big]\right) \approx \frac{k-1}{N-1}$. By substituting into $w$, we obtain $w \approx C'$.

Note that $-\alpha M(\pi;\mu)$ is exactly the return of KL-regularized MDP in Theorem 7. By the equivalence of this KL-regularized MDP and the KL-regularized AMG in (9), we have $M(\pi;\mu) = -\frac{\bar{J}(\pi;\mu)}{\alpha}$. Thus, minimization of (38) is equivalent to maximization of $\bar{J}(\pi;\mu)$.

## C.3 Full Expectation-Maximization Algorithm

In previous subsection, the reference policy $\mu$ is assumed as a prior, and the optimized policy would be constrained close to it through KL divergence. In practice, the prior of optimal policy can not

easily obtained, and a popular methodology to handle this is to learn the prior itself in the data-driven way, i.e., the principle of empirical Bayes.

The prior learning is done by maximizing the log-marginal likelihood:

$$L(\mu) = \log \mathbb{P}\left(\mathcal{O}_{0:L}\right) = \log \int \mathbb{P}\left(s_{0:L}, a_{0:L}, \tau_{0:L-1}, \mathcal{O}_{0:L}\right) ds_{0:L} da_{0:L} d\tau_{0:L-1}, \qquad (39)$$

where $\mathbb{P}\left(s_{0:L}, a_{0:L}, \tau_{0:L-1}, \mathcal{O}_{0:L}\right)$ is given in (33). As the log function includes a high-dimensional integration, evaluating $L(\mu)$ incurs intensive computation. Expectation-Maximization algorithm instead considers a lower bound of $L(\mu)$ to make the evaluation/optimization tractable:

$$L(\mu) \geq \log \mathbb{P}\left(\mathcal{O}_{0:L}\right) - D_{\mathrm{KL}}\left(\widehat{\mathbb{P}}\left(s_{0:L}, a_{0:L}, \tau_{0:L-1}\right) \,\Big|\Big|\, \mathbb{P}\left(s_{0:L}, a_{0:L}, \tau_{0:L-1} | \mathcal{O}_{0:L}\right)\right)$$

$$= \int \widehat{\mathbb{P}}\left(s_{0:L}, a_{0:L}, \tau_{0:L-1}\right) \log \mathbb{P}\left(s_{0:L}, a_{0:L}, \tau_{0:L-1}, \mathcal{O}_{0:L}\right) ds_{0:L} da_{0:L} d\tau_{0:L-1}$$

$$- \mathcal{H}\left[\widehat{\mathbb{P}}\left(s_{0:L}, a_{0:L}, \tau_{0:L-1}\right)\right], \qquad (40)$$

where the inequality is due to the non-negativity of KL divergence, and $\widehat{\mathbb{P}}\left(s_{0:L}, a_{0:L}, \tau_{0:L-1}\right)$ is an approximation to the exact posterior $\mathbb{P}\left(s_{0:L}, a_{0:L}, \tau_{0:L-1} | \mathcal{O}_{0:L}\right)$. The lower bound is tighter with the more exact approximation for the posterior. In previous subsection, we introduce the structured variational approximation with form of (35) to emphasize pessimism on the transition dynamics. Although this variational posterior would lead to non-zero KL term, it promotes learning robust policy as we discussed in previous subsection. Since that the variational posterior is with an adjustable policy $\pi$, we denote the lower bound by $\bar{L}(\mu; \pi)$.

By substituting (35) into (40), it follows

$$\bar{L}(\mu; \pi) = \mathbb{E}_{\rho_0, \widetilde{\mathbb{P}}_T, \tau_{0:L-1}, \pi}\left[\sum_{t=0}^{L} \log \mu(a_t | s_t)\right] + C''$$

$$= \mathbb{E}_{\rho_0, \widetilde{\mathbb{P}}_T, \tau_{0:L-1}, \pi}\left[\sum_{t=0}^{L} \log \mu(a_t | s_t) - \log \pi(a_t | s_t) + \log \pi(a_t | s_t)\right] + C''$$

$$= \mathbb{E}_{\rho_0, \widetilde{\mathbb{P}}_T, \tau_{0:L-1}, \pi}\left[\sum_{t=0}^{L} -D_{\mathrm{KL}}(\pi(\cdot | s_t) \,||\, \mu(\cdot | s_t))\right] + C''', \qquad (41)$$

where $C''$ and $C'''$ includes the constant terms irrelevant to $\mu$. According to the form of (41), given fixed $\pi$, the optimal prior policy to maximize $\bar{L}(\mu; \pi)$ is obtained as $\mu = \pi$. Maximizing the lower bound is known as Maximization step.

Recall $\pi$ in the variational posterior is adjustable, we can optimize it by minimizing $D_{\mathrm{KL}}\left(\widehat{\mathbb{P}}\left(s_{0:L}, a_{0:L}, \tau_{0:L-1}\right) \,\Big|\Big|\, \mathbb{P}\left(s_{0:L}, a_{0:L}, \tau_{0:L-1} | \mathcal{O}_{0:L}\right)\right)$ to tighten the bound. The minimization procedure is known as Expectation step. In our case, the minimization problem is exactly the one discussed in previous subsection.

When repeatedly and alternately applying the Expectation and Maximization steps, the iterative regularized policy optimization algorithm is recovered. According to Theorem 5, both $\pi$ and $\mu$ continuously improve regarding the objective function.

# D  Algorithm and Implementation Details for Model-Based Offline RL with PMDB

The pseudocode for model-based offline RL with PMDB is presented in Algorithm 1. As $\rho_0$ is unknown in practice, we uniformly sample states from $\mathcal{D}$ as the initial $\{s_0\}$. In Step 4, the primary players act according to the non-parametric policy $\pi$, rather than its approximated policy $\pi_\phi$. This is because during learning process $\pi_\phi$ is not always trained adequately to approximate $\pi$, then following $\pi_\phi$ will visit unexpected states. In Step 11, the reference policy $\pi_{\phi'}$ is returned as the final policy, considering that it is more stable than $\pi_\phi$.

**Algorithm 1** Model-Based Offline RL with PMDB
___
**Require**: $\mathcal{D}, \mathbb{P}_T, N, k, M$.

1: **Approximator initialization:** Randomly initialize Q-function $Q_\theta(s, a)$ and policy $\pi_\phi(a|s)$; Initialize target Q-function $Q_{\theta'}(s, a)$ and reference policy $\pi_{\phi'}(a|s)$ with $\theta' \leftarrow \theta, \phi' \leftarrow \phi$.
2: **Game initialization:** Randomly sample $C$ states from $\mathcal{D}$, as the initial states for $C$ paralleled games $\{s\}$.
3: **for** step $t = 1, 2, \cdots, M$ **do**
4:    **Primary players:** Sample actions according to

$$\pi(a|s) \propto \pi_{\phi'}(a|s) \exp\left(\frac{1}{\alpha} Q_\theta(s, a)\right).$$

5:    **Game transitions:** Sample candidate sets $\{\mathcal{T}\}$ according to (3).
6:    **Update:** Sample a batch of transitions from $\mathcal{D}$, together with the $C$-size game transitions $\{(s, a, \mathcal{T})\}$, to update $\theta$ and $\phi$ via one-step gradient descent regarding (11) and (14).
7:    **Secondary players:** Determine whether to exploit or explore: with probability of $(1 - \epsilon)$,

$$\bar{\tau} = \arg\lfloor\min\rfloor_{\tau \in \mathcal{T}}^{k} \mathbb{E}_\tau \left[\alpha \log \mathbb{E}_{\pi_{\phi'}} \exp\left(\frac{1}{\alpha} Q_\theta(s', a')\right)\right],$$

   otherwise randomly choose $\bar{\tau}$ from $\mathcal{T}$.
8:    **Game transitions:** Sample states following $\{\bar{\tau}\}$ to update $\{s\}$. For terminal states in $\{s\}$, use random samples from $\mathcal{D}$ to replace them.
9:    **Moving-average update:** Update reference policy and target Q-function with

$$\phi' \leftarrow \omega_1 \phi + (1 - \omega_1)\phi',$$
$$\theta' \leftarrow \omega_2 \theta + (1 - \omega_2)\theta'.$$

10: **end for**
11: **return** $\pi_{\phi'}$.
___

**Computing expectation** Algorithm 1 involves the computation of expectation. In discrete domains, the expectation can be computed exactly. In continuous domains, we use Monte Carlo methods to approximate it. Concretely, for the expectation over states we apply vanilla Monte Carlo sampling, while for the expectation over actions we apply importance sampling. To elaborate, the expectation over actions can be written as

$$\mathbb{E}_\mu \exp\left(\frac{1}{\alpha} Q(s, a)\right) = \frac{1}{2}\left[\mathbb{E}_\mu \exp\left(\frac{1}{\alpha} Q(s, a)\right) + \mathbb{E}_q \frac{\mu(a|s) \exp\left(\frac{1}{\alpha} Q(s, a)\right)}{q(a|s)}\right]$$

$$\approx \frac{1}{2n}\left[\sum_{a_i \sim \mu(\cdot|s)}^{n} \exp\left(\frac{1}{\alpha} Q(s, a_i)\right) + \sum_{a_i \sim q(\cdot|s)}^{n} \frac{\mu(a_i|s) \exp\left(\frac{1}{\alpha} Q(s, a_i)\right)}{q(a_i|s)}\right],$$

where $q$ is the proposal distribution.

In Algorithm 1, the above expectation is computed for both $s \in \mathcal{D}$ and $s \in \mathcal{D}'$. For $s \in \mathcal{D}$, we choose

$$q(\cdot|s) = \mathcal{N}(\,\cdot\,; a, \sigma^2 I), \quad \text{where } a|s \sim \mathcal{D},$$

i.e., the samples are drawn close to the data points, and $\sigma^2$ determines how much they keep close. For example, in Step 6 a batch of $\{(s, a, s')\}$ are sampled from $\mathcal{D}$ to calculate (14), then we construct the proposal distribution as above for each $(s, a)$ in the batch. The motivation of drawing actions near the data samples is to enhance learning in the multi-modal scenario, where the offline dataset $\mathcal{D}$ is collected by mixture of multiple policies. If $\mu$ is single-modal (say the widely adopted Gaussian policy) and we solely draw samples from it to approximate the expectation, these samples will be locally clustered. Then, applying them to update $\pi_\theta$ in (14) can be easily get stuck at local optimum.

For $s \in \mathcal{D}'$, we choose

$$q(\cdot|s) = \pi_\theta(\cdot|s).$$

The reason is that $\pi_\theta$ is an approximator to the improved policy with higher Q-value, and sampling from it hopefully reduces variance of the Monte Carlo estimator.

Although applying Monte Carlo methods to approximate the expectation incurs extra computation, all the operators can be executed in parallel. In the experiments, we use 10 and 20 samples respectively for the expectations over state and action, and the algorithm is run on a single machine with one Quadro RTX 6000 GPU. The results show that in average it takes 73.4 s to finish 1k training steps, and the GPU memory cost is 2.5 GB.

Several future directions regarding the Monte Carlo method are worthy to explore. For example, by reducing the sample size for the expectation over state, the optimized policy additionally tends to avoid the risk due to aleatoric uncertainty (while in this work we focus on epistemic uncertainty). Besides, the computational cost can be reduced by more aggressive Monte Carlo approximation, for example only using mean action to compute the expectation in terms of policy. We leave these as future work.

## E   Choice of Initial Dynamics Belief

In offline setting, extra knowledge is strongly desired to aggressively optimize policy. The initial dynamics belief provides an interface to absorb the aforehand knowledge of system transition. In what follows, we illustrate several potential usecases:

- Consider the physical system where the dynamics can be described as mathematical expression but with uncertain parameter. If we have a narrow distribution over the parameter (according to expert knowledge or inferred from data), the system is almost known for certain. Here, both the mathematical expression and narrow distribution provide more information.

- Consider the case where we know the dynamics is smooth with probability of 0.7 and periodic with probability of 0.3. Gaussian processes (GPs) with RBF kernel and periodic kernel can well encode these prior knowledge. Then, the 0.7-0.3 mixture of the two GPs trained with offline data can act as the dynamics belief to provide more information.

- In the case where multi-task datasets are available, we can train dynamics models using each of the datasets and assign likelihood ratios to these models. If the likelihood ratio well reflects the similarity between the concerned task and the offline tasks, the multi-task datasets promote knowledge.

The performance gain is expected to monotonously increase with the amount of correct knowledge. As an impractical but intuitive example, with the exact knowledge of system transition (the initial belief is a delta function), the proposed approach is actually optimizing policy as in real system.

In practice, the expert knowledge is not available everywhere. When unavailable, the best we can hope for is that the final policy stays close to the dataset, but unnecessary to be fully covered (as we want to utilize the generalization ability of dynamics model at least around the data). To that end, the dynamics belief is desired to be certain at the region in distribution of dataset, and turns more and more uncertain when departing. It has been reported that the simple model ensemble leads to such a behavior [12]. In this sense, the uniform distribution over learned dynamics ensemble can act as a quite common belief. In the experiments, we apply it for fair comparison with baseline methods.

## F   Automatically Adjusting KL Coefficient

In Section 4, the KL regularizer is introduced to restrict $\pi_\phi$ in a small region near $\pi_{\phi'}$, such that the Q-value can be evaluated sufficiently before policy improvement. Apart from fixing the KL coefficient $\alpha$ throughout, we provide a strategy to automatically adjust it.

Note that the optimal policy to minimize $L_P$ in (14) is $\frac{\pi_{\phi'}(\cdot|s)\exp\left(\frac{1}{\alpha}Q_\theta(s,\cdot)\right)}{\mathbb{E}_{\pi_{\phi'}}\left[\exp\left(\frac{1}{\alpha}Q_\theta(s,a)\right)\right]}$. The criterion of choosing $\alpha$ is to constrain the KL divergence between this policy and $\pi_{\phi'}$ smaller than a specified constant, i.e.,

$$D_{\mathrm{KL}}\left(\frac{\pi_{\phi'}(\cdot|s)\exp\left(\frac{1}{\alpha}Q_\theta(s,\cdot)\right)}{\mathbb{E}_{\pi_{\phi'}}\left[\exp\left(\frac{1}{\alpha}Q_\theta(s,a)\right)\right]}\,\middle\|\,\pi_{\phi'}(\cdot|s)\right) \leq d. \tag{42}$$

Finding $\alpha$ to satisfy the above inequation is intractable, instead we consider a surrogate of the KL divergence:

$$D_{\mathrm{KL}}\left(\frac{\pi_{\phi'}(\cdot|s)\exp\left(\frac{1}{\alpha}Q_\theta(s,\cdot)\right)}{\mathbb{E}_{\pi_{\phi'}}\left[\exp\left(\frac{1}{\alpha}Q_\theta(s,a)\right)\right]}\ \middle\|\ \pi_{\phi'}(\cdot|s)\right)$$

$$= \mathbb{E}_{\pi_{\phi'}}\left[\frac{\exp\left(\frac{1}{\alpha}Q_\theta(s,a)\right)}{\mathbb{E}_{\pi_{\phi'}}\left[\exp\left(\frac{1}{\alpha}Q_\theta(s,a)\right)\right]}\cdot\frac{1}{\alpha}Q_\theta(s,a)\right] - \log\mathbb{E}_{\pi_{\phi'}}\left[\exp\left(\frac{1}{\alpha}Q_\theta(s,a)\right)\right]$$

$$\leq \frac{1}{\alpha}\left(\mathbb{E}_{\pi_{\phi'}}\left[\frac{\exp\left(\frac{1}{\alpha_0}Q_\theta(s,a)\right)}{\mathbb{E}_{\pi_{\phi'}}\left[\exp\left(\frac{1}{\alpha_0}Q_\theta(s,a)\right)\right]}\cdot Q_\theta(s,a)\right] - \mathbb{E}_{\pi_{\phi'}}\left[Q_\theta(s,a)\right]\right),$$

where $\alpha_0$ is a predefined lower bound of $\alpha$.

Then, (42) can be satisfied by setting

$$\alpha \geq \frac{1}{d}\left(\mathbb{E}_{\pi_{\phi'}}\left[\frac{\exp\left(\frac{1}{\alpha_0}Q_\theta(s,a)\right)}{\mathbb{E}_{\pi_{\phi'}}\left[\exp\left(\frac{1}{\alpha_0}Q_\theta(s,a)\right)\right]}\cdot Q_\theta(s,a)\right] - \mathbb{E}_{\pi_{\phi'}}\left[Q_\theta(s,a)\right]\right).$$

Combining with the predefined lower bound, we choose $\alpha$ as

$$\alpha = \max\left(\frac{1}{d}\left(\mathbb{E}_{\pi_{\phi'}}\left[\frac{\exp\left(\frac{1}{\alpha_0}Q_\theta(s,a)\right)}{\mathbb{E}_{\pi_{\phi'}}\left[\exp\left(\frac{1}{\alpha_0}Q_\theta(s,a)\right)\right]}\cdot Q_\theta(s,a)\right] - \mathbb{E}_{\pi_{\phi'}}\left[Q_\theta(s,a)\right]\right), \alpha_0\right).$$

In practice, the expectation can be estimated over Monte Carlo samples. Note that the coefficient can be computed individually for each state, picking $d$ is hopefully easier than picking $\alpha$ suitable for all states.

## G  Additional Experimental Setup

**Task Domains**   We evaluate the proposed methods and the baselines on eighteen domains involving three environments (hopper, walker2d, halfcheetah), each with six dataset types. The dataset types are collected by different policies, denoted by *random*: a randomly initialized policy, *expert*: a policy trained to completion with SAC, *medium*: a policy trained to approximately 1/3 the performance of the expert, *medium-expert*: 50-50 mixture of medium and expert data, *medium-replay*: the replay buffer of a policy trained up to the performance of the medium agent, *full-replay*: the replay buffer of a policy trained up to the performance of the expert agent.

**Dynamics Belief**   We adopt an uniform distribution over dynamics model ensemble as the initial belief. The ensemble contains 100 neural networks, each is with 4 hidden layers and 256 hidden units per layer. All the neural networks are trained independently with the sample dataset $\mathcal{D}$ and in parallel. The training process stops after the average training loss does not change obviously. Specifically, the number of epochs for hopper-random and walker2d-medium are 2000, and those for other tasks are 1000. Note that the level of pessimism depends on the candidate size $N$ ($= 10$ by default), rather than the ensemble size.

**Policy Network and Q Network**   The policy network is with 3 hidden layers and 256 hidden units per layer. It outputs the mean and the diagonal variance for a Gaussian distribution, which is then transformed via tanh function to generate the policy. When evaluating our approach, we apply the deterministic policy, where the action is the tanh transformation of the Gaussian mean. The Q network is with the same architecture as the policy network except the output layer. Similar to existing RL approaches [35], we make use of two Q networks and apply the minimum of them for calculation in Algorithm 1, in order to mitigate over-estimation when learning in the AMG. The policy learning stops after the performance in AMG does not change obviously. Specifically, the gradient steps for walker2d-random, halfcheetah-random and hopper with all dataset types are 1 million, and those for other tasks are 2 millon.

**Hyperparameters**  We list the detailed hyperparameters in Table 3.

| Parameter | Value |
|---|---|
| dynamics learning rate | $10^{-4}$ |
| policy learning rate | $3 \cdot 10^{-5}$ |
| Q-value learning rate | $3 \cdot 10^{-4}$ |
| discounted factor ($\gamma$) | 0.99 |
| smoothing coefficient for policy ($\omega_1$) | $10^{-5}$ |
| smoothing coefficient for Q-value ($\omega_2$) | $5 \cdot 10^{-3}$ |
| Exploration ratio for secondary player ($\epsilon$) | 0.1 |
| KL coefficient ($\alpha$) | 0.1 |
| variance for important sampling ($\sigma^2$) | 0.01 |
| Batch size for dynamics learning | 256 |
| Batch size for AMG and MDP | 128 |
| Maximal horizon of AMG | 1000 |

Table 3: Hyperparameters

## H  Practical Impact of $N$

Table 4 lists the impact of $N$. The performance in the AMGs improve when decreasing $k$. Regarding the performance in true MDPs, we notice that $N = 15$ corresponds to the best performance for hopper, but for the others $N = 5$ is better.

| | hopper-medium | | walker2d-medium | | halfcheetah-medium | |
|---|---|---|---|---|---|---|
| $N$ | MDP | AMG | MDP | AMG | MDP | AMG |
| 5 | 90.2±25.4 | 108.6±2.2 | 112.7±0.9 | 101.7±5.7 | 79.8±0.4 | 69.5±1.6 |
| 10 | 106.8±0.2 | 105.2±1.6 | 94.2±1.1 | 77.2±3.7 | 75.6±1.3 | 67.3±1.1 |
| 15 | 107.3±0.2 | 103.1±1.8 | 92.1±0.3 | 68.3±6.7 | 75.4±0.4 | 63.2±2.3 |

Table 4: Impact of $N$, with $k = 2$.

## I  Ablation of Randomness of $\mathcal{T}$

Compared to the standard Bellman backup operator in Q-learning, the proposed one additionally includes the expectation over $\mathcal{T} \sim \mathcal{P}_T^N$ and the $k$-minimum operator over $\tau \in \mathcal{T}$. We report the impact of choosing different $k$ in Table 2, and present the impact of the randomness of $\mathcal{T}$ as below. Fixed $\mathcal{T}$ denotes that after sampling once $\mathcal{T}$ from the belief distribution we keep it fixed during policy optimization.

| Task Name | Stochastic $\mathcal{T}$ | Fixed $\mathcal{T}$ |
|---|---|---|
| hopper-medium | $106.8 \pm 0.2$ | $106.2 \pm 0.3$ |
| walker2d-medium | $94.2 \pm 1.1$ | $90.1 \pm 4.3$ |
| halfcheetah-medium | $75.6 \pm 1.3$ | $73.1 \pm 2.8$ |

Table 5: Impact of randomness of $\mathcal{T}$

We observe that the randomness of $\mathcal{T}$ has a mild effect on the performance in average. The reason can be that we apply the uniform distribution over dynamics ensemble as initial belief (without additional knowledge to insert). The model ensemble is reported to produce low uncertainty estimation in distribution of data coverage and high estimation when departing the dataset [12]. This property makes the optimized policy keep close to the dataset, and it does not rely on the randomness of ensemble elements. However, involving the randomness can lead to more smooth variation of the estimated uncertainty, which benefits the training process and results in better performance. Apart from these empirical results, we highlight that in cases with more informative dynamics belief, only picking several fixed samples from the belief distribution as $\mathcal{T}$ will result in the loss of knowledge.

## J    Weighting AMG Loss and MDP Loss in (11)

In (11), the Q-function is trained to minimize the Bellman residuals of both the AMG and the empirical MDP, equipped with the same weight (both are 1). In the following table, we show experiment results to check the impact of different weights.

| Task Name | 0.5:1.5 | 1.0:1.0 | 1.5:0.5 |
|---|---|---|---|
| hopper-medium | $106.6 \pm 0.3$ | $106.8 \pm 0.2$ | $106.5 \pm 0.3$ |
| walker2d-medium | $93.8 \pm 1.5$ | $94.2 \pm 1.1$ | $93.1 \pm 1.3$ |
| halfcheetah-medium | $75.2 \pm 0.8$ | $75.6 \pm 1.3$ | $76.1 \pm 1.0$ |

Table 6: Impact of weights in (11)

The results suggests that the performance does not obviously depend on the weights. But in cases with available expert knowledge about dynamics, the weights can be adjusted to match our confidence on the knowledge, i.e., the less confidence, the smaller weight for AMG.

## K    Comparison with RAMBO

We additionally compared the proposed approach with RAMBO [61], a concurrent work that also formulates offline RL as a two-player zero-sum game. The results of RAMBO for random, medium, medium-expert and medium-replay are taken from [61]. For the other two dataset types, we run the official code and follow the hyperparameter search procedure reported in its paper.

| Task Name | BC | BEAR | BRAC | CQL | MOReL | EDAC | RAMBO | PMDB |
|---|---|---|---|---|---|---|---|---|
| hopper-random | 3.7±0.6 | 3.6±3.6 | 8.1±0.6 | 5.3±0.6 | **38.1±10.1** | 25.3±10.4 | 25.4±7.5 | 32.7±0.1 |
| hopper-medium | 54.1±3.8 | 55.3±3.2 | 77.8±6.1 | 61.9±6.4 | 84.0±17.0 | 101.6±0.6 | 87.0±15.4 | **106.8±0.2** |
| hopper-expert | 107.7±9.7 | 39.4±20.5 | 78.1±52.6 | 106.5±9.1 | 80.4±34.9 | **110.1±0.1** | 50.0±8.1 | **111.7±0.3** |
| hopper-medium-expert | 53.9±4.7 | 66.2±8.5 | 81.3±8.0 | 96.9±15.1 | 105.6±8.2 | **110.7±0.1** | 88.2±20.5 | **111.8±0.6** |
| hopper-medium-replay | 16.6±4.8 | 57.7±16.5 | 62.7±30.4 | 86.3±7.3 | 81.8±17.0 | 101.0±0.5 | 99.5±4.8 | **106.2±0.6** |
| hopper-full-replay | 19.9±12.9 | 54.0±24.0 | 107.4±0.5 | 101.9±0.6 | 94.4±20.5 | 105.4±0.7 | 105.2 ±2.1 | **109.1±0.2** |
| walker2d-random | 1.3±0.1 | 4.3±1.2 | 1.3±1.4 | 5.4±1.7 | 16.0±7.7 | 16.6±7.0 | 0.0±0.3 | **21.8±0.1** |
| walker2d-medium | 70.9±11.0 | 59.8±40.0 | 59.7±39.9 | 79.5±3.2 | 72.8±11.9 | 92.5±0.8 | 84.9 ±2.6 | **94.2±1.1** |
| walker2d-expert | 108.7±0.2 | 110.1±0.6 | 55.2±62.2 | 109.3±0.1 | 62.6±29.9 | **115.1±1.9** | 1.6±2.3 | **115.9±1.9** |
| walker2d-medium-expert | 90.1±13.2 | 107.0±2.9 | 9.3±18.9 | 109.1±0.2 | 107.5±5.6 | **114.7±0.9** | 56.7±39.0 | 111.9±0.2 |
| walker2d-medium-replay | 20.3±9.8 | 12.2±4.7 | 40.1±47.9 | 76.8±10.0 | 40.8±20.4 | 87.1±2.3 | **89.2±6.7** | 79.9±0.2 |
| walker2d-full-replay | 68.8±17.7 | 79.6±15.6 | 96.9±2.2 | 94.2±1.9 | 84.8±13.1 | **99.8±0.7** | 88.3±4.9 | 95.4±0.7 |
| halfcheetah-random | 2.2±0.0 | 12.6±1.0 | 24.3±0.7 | 31.3±3.5 | **38.9±1.8** | 28.4±1.0 | **39.5±3.5** | 37.8 ± 0.2 |
| halfcheetah-medium | 43.2±0.6 | 42.8±0.1 | 51.9±0.3 | 46.9±0.4 | 60.7±4.4 | 65.9±0.6 | **77.9 ±4.0** | 75.6± 1.3 |
| halfcheetah-expert | 91.8±1.5 | 92.6±0.6 | 39.0±13.8 | 97.3±1.1 | 8.4±11.8 | 106.8±3.4 | 79.3±15.1 | **105.7± 1.0** |
| halfcheetah-medium-expert | 44.0±1.6 | 45.7±4.2 | 52.3±0.1 | 95.0±1.4 | 80.4±11.7 | 106.3±1.9 | 95.4 ±5.4 | **108.5±0.5** |
| halfcheetah-medium-replay | 37.6±2.1 | 39.4±0.8 | 48.6±0.4 | 45.3±0.3 | 44.5±5.6 | 61.3±1.9 | 68.7± 5.3 | **71.7±1.1** |
| halfcheetah-full-replay | 62.9±0.8 | 60.1±3.2 | 78.0±0.7 | 76.9±0.9 | 70.1±5.1 | 84.6±0.9 | 87.0±3.2 | **90.0±0.8** |
| Average | 49.9 | 52.4 | 54.0 | 73.7 | 65.1 | 85.2 | 68.0 | **88.2** |

Table 7: Extended Results for D4RL datasets.

The results show our approach outperforms RAMBO on most of considered tasks. One reason can be that the problem formulation of RAMBO is based on robust MDP, whose defects are discussed in Section 2 and Appendix A.