# OpenReview forum: "Model-Based Offline Reinforcement Learning with Pessimism-Modulated Dynamics Belief"
_NeurIPS.cc/2022/Conference — NeurIPS 2022 Accept_

### Official Review · Reviewer_998y · 2022-07-09

**Rating:** 8
**Confidence:** 4
**Soundness:** 4 excellent
**Presentation:** 3 good
**Contribution:** 4 excellent

**Summary:**

Existing studies on model-based offline RL often rely on the quantification of the  uncertainty in the learned dynamics and utilize it to regularize the reward.  One drawback of this approach  is that it may result in unreliable evaluation on the impact of the uncertainty and incur unexpected tradeoff between model utilization and risk avoidance. To resolve this issue, the authors introduced a novel model-based offline RL algorithm with pessimism-modulated dynamics belief, where the uncertainty is not explicitly quantified but  a belief distribution over system dynamics is used for sampling instead, when updating the policy evaluation and policy update.  The sampling procedure is cleverly modeled as an alternating Markov game and the degree of pessimism can be determined by the hyper parameters in the sampling procedure. The proposed offline RL algorithm solves the AMG by iterative regularized policy optimization with monotonous improvement guarantee. The empirical results on D4RL shows the efficiency and performance improvement of the proposed algorithm.

**Questions:**

How to construct a sufficient belief without expert knowledge  for an arbitrary task? How to tune the hyper parameter $k$ and $N$ when processing a new task?

How to determine the weights in Eqn. (11)? The optimization over the Bellman residual of both the AMF and the empirical MDP ia applied with the same weight (both are 1).

**Limitations:**

One limitation of this work is that is is unclear a prior how to obtain a sufficient belief without expert knowledge. It will be of interest to shed light on how to construct a valuable (initial) dynamic belief for arbitrary tasks.

**Strengths And Weaknesses:**

Strength:
+ The proposed methodology is explained in detail and the paper is easy to follow. The empirical results and ablation studies shwocase the efficiency of the proposed approach on the standard benchmarks.

+ The formulation of offline RL as AMG is interesting and leads to the development of pessimism-modulated dynamics belief derivation.

+ The theoretical results on the pessimism-modulated dynamics belief is promising and serves good guidance on the empirical offline RL algorithm design.

Weakness:
- The experimental results can be made more comprehensive. In Figures 2 and 3, what is the possible cause of the large dip in half-cheetah-medium large dip in halfcheetah-medium task?

- How to tune the hyper parameter $k$ and $N$ when processing a new task?

- How to determine the weights in Eqn. (11)? The optimization over the Bellman residual of both the AMF and the empirical MDP ia applied with the same weight (both are 1).

- Line 194,   the reference to Eqn. (18) is undefined.

---

> ### Author Response · Authors · 2022-08-02
> **Official Response to Reviewer 998y**
>
> Thanks for your insightful comments. We provide clarification to your questions and concerns as below.
>
> **Q1: How to construct a sufficient belief without expert knowledge for an arbitrary task?**
>
> A1: In cases where expert knowledge is unavailable, the best we can hope for is that the final policy stays close to the dataset, but unnecessary to be fully covered (as we want to utilize the generalization ability of dynamics model at least around the data). To that end, the dynamics belief is desired to be certain at the region in distribution of dataset, and turns more and more uncertain when departing. It has been reported that the simple model ensemble leads to such a behavior, for example Figure 1 in [1]. In this sense, the uniform distribution over learned dynamics ensemble can act as a quite common belief. For small dataset, Gaussian process with smooth kernel is also a good choice to achieve this. We will add such explanation in the paper.
>
> [1] Lakshminarayanan, B., et al. Simple and scalable predictive uncertainty estimation using deep ensembles. NeurIPS 2017.
>
> **Q2: How to tune the hyper parameter $k$ and $N$ when processing a new task?**
>
> A2: The performance in AMG can be treated as surrogate to tune $k$ and $N$ offline. To be concrete, we set an anticipation on how the performance is after offline optimization, according to the statistics of the per-trajectory returns contained by the dataset, or other knowledge of the task and dataset. With the monotonicities of $k$ and $N$, they can be tuned such that the achieved performance in AMG is close to the anticipated performance. As we have a reasonable anticipation, the performance of AMG hopefully serves as a good surrogate to the real performance, like that in Figure 2, Tables 2 and 4.
>
> We believe that the hyper-parameter tuning without online access is challenging for any offline algorithm. Apart from introducing offline surrogate, the robustness of hyper-parameter is also crucial. In this paper, we empirically verify the robustness of $k$ and $N$, i.e., the results in Table 1 keeps the same $k$ and $N$ over all tasks, changing $k$ and $N$ still retains comparable/competitive performance in Tables 2 and 4. Concurrently, the algorithmic robustness is also explored in the ICML'22 outstanding paper [2].
>
> [2] Cheng, C. A., et al. Adversarially trained actor critic for offline reinforcement learning. ICML 22.
>
> **Q3: How to determine the weights in Eqn. (11)? The optimization over the Bellman residual of both the AMF and the empirical MDP is applied with the same weight (both are 1).**
>
> A3: We add experiments to check the impact of different weights:
> Task Name|0.5:1.5| 1.0:1.0 |1.5:0.5
> -|-|-|-
> hopper-medium | 106.6$\pm$0.3 | 106.8$\pm$0.2 | 106.5$\pm$0.3
> walker2d-medium| 93.8$\pm$1.5   | 94.2$\pm$1.1 | 93.1$\pm$1.3
> halfcheetah-medium| 75.2$\pm$0.8   | 75.6$\pm$1.3 | 76.1$\pm$1.0
>
> In the above, the performance does not obviously depend on the weights. But in cases where expert knowledge about dynamics is available, we would like to insert it into dynamics belief. The weights can be adjusted to match our confidence on the knowledge, i.e., the less confidence, the smaller weight for AMG.
>
> **Weakness: The experimental results can be made more comprehensive. In Figures 2 and 3, what is the possible cause of the large dip in half-cheetah-medium large dip in halfcheetah-medium task?**
>
> A4: We will add the experimental results mentioned in A3, [A3 to Reviewer y5WY] and [A4 to Reviewer LbFj] to the main paper.
>
> In Figure 3, we notice that the large dip of Q-value is accompanied with the sudden raise of dynamics uncertainty. We suspect this is due to the optimized policy being far from the dataset. We try to verify by checking the maximal return covered by the offline dataset. It shows that the maximal normalized returns provided by offline datasets are 99.6, 92.0 and 45.0 respectively for hopper, walker2d and halfcheetah, while the proposed approach achieves 106.8, 94.2 and 75.6. The policy optimization is more significant for halfcheetah (where we observe the large dip), indicating the policy should move further from the dataset.
>
> The above finding also explains why the AMG performance in Figure 3 runs into large dip only for halfcheetah: with the larger dynamics uncertainty, the secondary player can choose the more pessimistic transition. However, we want to highlight that it is normal behavior of the proposed algorithm and does not mean instability, as we are handling the alternating Markov game, a specialization of zero-sum game. Besides, we can see that even when the AMG performance goes down the MDP performance is still stable.
>
> **Weakness: Eqn. (18) undefined**
>
> A5: (18) will be replaced by (7). Thanks for pointing it out.

---

> > ### Comment · Reviewer_998y · 2022-08-08
> > **Congrats on your nice results**
> >
> > Your response to my comments is reasonable and objective.  Congrats on a nice paper!

---

### Official Review · Reviewer_Bm2w · 2022-07-11

**Rating:** 7
**Confidence:** 3
**Soundness:** 3 good
**Presentation:** 2 fair
**Contribution:** 3 good

**Summary:**

The paper proposes a mechanism for offline RL based on a two player game with a Bayesian environment, in which an RL actor picks a policy and a environment actor picks environment transitions from a likely set pessimistically. The authors show general convergence and existence results in their framework and empirical results based on the established D4RL framework.

**Questions:**

- 295: if you assume online-access here, do you break offline RL assumptions? In that case, the core assumptions of the scenario should be clarified here. This would make the comparison between this algorithm and fully offline methods slightly less convincing, if you assume i.e. a few-shot scenario.
- Theorem 2: is that MDP stationary? It depends on pi, so that might make things more complicated? If the policy has to be learned in a changing MDP, how does that affect learning? Theorem 1 states that the operator is a contractive mapping, but
- (14) why not KL as difference of distributions, or Wasserstein? Is there a specific reason for this concrete formulation?

**Limitations:**

A brief discussion is given, without mentioning any major drawbacks in depth.

**Strengths And Weaknesses:**

== Strengths ==
- The core theory is well laid out and supported with theoretical results as well as empirical investigation.
- The structure is overall well laid out and easy to follow. Some minor points of confusion and suggestions are discussed below, but in general, the paper is presented in a clear legible fashion.

== Weaknesses ==
- The paper is missing a clear related work section. This makes it harder to position it in the relevant literature and makes it difficult to assess whether concurrent similar strands of the literature were reviewed. Since I am not an expert in the concrete field the authors are tackling, some of the following concerns might be wrong or missing relevant details, but I will list them for discussion:
  - My biggest concern is that no other works on Bayesian Robust RL were cited. The formulation the authors present seems novel, but to the best of my knowledge several other works have tackled Bayesian Robust RL and a clear comparison would greatly improve the paper. I.e. how does the proposed method relate to https://arxiv.org/abs/1905.08188 and similar works.
  - The other strand of research I am missing comparison to is CVaR style algorithms, which seem similar in its setup in targeting a quantile of returns. For example, this paper from NeurIPS 2021 seems very related https://papers.nips.cc/paper/2021/hash/08f90c1a417155361a5c4b8d297e0d78-Abstract.html
- The set generating mechanism and definitions in (3) could be explained better. Big $\tau$ seems to refer to a set of transitions, but $\bar{s}$ is a state (or a single transition)? This seems to be explained in the text under the equations, I think it might be clearer for readers if the state and action spaces of the 2nd player was fully explained in the introduction of the formalism.

Unclear lines/grammatical errors:
- Line 51: with entirely different implication compared to the reward or Q-value?
- Line 140: To differentiate with uncertainty set we call it candidate set, but to avoid redundancy we reuse the notation of T?
- Line 24: in vast of applications?

---

> ### Author Response · Authors · 2022-08-02
> **Official Response to Reviewer Bm2w (part 2/2)**
>
> **Weakness: The set generating mechanism and definitions in (3) could be explained better. $\mathcal{T}$ seems to refer to a set of transitions, but $\bar{s}$ is a state (or a single transition)? This seems to be explained in the text under the equations, I think it might be clearer for readers if the state and action spaces of the 2nd player was fully explained in the introduction of the formalism.**
>
> A5: $\bar{s}$ is the state for secondary player in general AMG. After introducing the general AMG, we instantiate it in the setting of offline RL, where $\bar{s}=\mathcal{T}$ means that the state of secondary player is the set of plausible transitions. Thanks for your suggestion, we will try best to improve.
>
> **Unclear lines/grammatical errors**
>
> A6: We will fix them in the paper.
>
> **Limitations: A brief discussion is given, without mentioning any major drawbacks in depth.**
>
> A7: The initial belief provides the interface to insert additional knowledge, but also introduces potential risks: 1) when inserting incorrect or biased knowledge, the optimization procedure would be misled and the reality gap can be amplified; 2) when considering data-driven approach to learn extra knowledge from multi-task datasets, it is not straightforward to devise a principled criterion on similarity measurement between the concerned task and the multiple tasks. This brings challenge to insert knowledge for arbitrary tasks. However, with only offline dataset, it is always expected to integrate other knowledge to obtain significant policy improvement. We hope highlighting these inspires more research on this topic.

---

> > ### Comment · Reviewer_Bm2w · 2022-08-07
> > **Thanks for the through answer**
> >
> > Dear authors, thanks for the thorough answer to my concerns. I think all of my questions were answered, I will  update my score accordingly an recommend acceptance.

---

> > > ### Author Response · Authors · 2022-08-08
> > > **Response to Reviewer Bm2w**
> > >
> > > We are glad to have addressed your concern. Thanks for the apprecation of this work and raising the score!

---

> ### Author Response · Authors · 2022-08-02
> **Official Response to Reviewer Bm2w (part 1/2)**
>
> Thanks for your constructive comments. We provide clarification to your questions and concerns as below.
>
> **Q1: 295: if you assume online-access here, do you break offline RL assumptions? In that case, the core assumptions of the scenario should be clarified here. This would make the comparison between this algorithm and fully offline methods slightly less convincing, if you assume i.e. a few-shot scenario.**
>
> A1: The on-policy data is collected in the constructed AMG, thus no assumption of online access is needed. In experiment, the proposed algorithm is examined with solely using the offline data. We will revise the misleading description in the paper.
>
> **Q2: Theorem 2: is that MDP stationary? It depends on pi, so that might make things more complicated? If the policy has to be learned in a changing MDP, how does that affect learning? Theorem 1 states that the operator is a contractive mapping, but**
>
> A2: The equivalent MDP depends on $\pi$ and indeed changes during policy optimization. Theorem 2 is from view of connecting the proposed formulation with the original problem, but not from the algorithmic view. When dealing with policy optimization in Section 4, we do not rely on the equivalent MDP to devise algorithm. Theorem 1 is about policy evaluation and considers fixed policy, while Theorems 4 and 5 are about policy optimization. They show that the policy improvement is convergent, and with the converged policy we can recall Theorem 2 to obtain the equivalent MDP.
>
> **Q3: (14) why not KL as difference of distributions, or Wasserstein? Is there a specific reason for this concrete formulation?**
>
> A3: We intentionally choose KL as the similarity measure between probabilities. The reason is that it can be tractably and unbiasedly estimated via Monte Carlo sampling $a\sim\pi_{\phi'}$. It would be interesting to explore other possible choices, such as Wasserstein distance or total variation distance, but we want to leave it for other works.
>
> **Weakness: Related works**
>
> A4: We will move the related works in Appendix A to the main paper, and also include the following discussion.
> - Bayesian Robust RL
>    * Bayesian Robust RL [1] is based on the problem setting of Bayesian RL, where new observations are continually received and utilized to make better decision. The goal of Bayesian RL is to fast **explore and adapt** when deployed in the environment which is pre-considered during training process. As comparison, offline RL focuses on how to sufficiently **exploit** the offline dataset to generate the best-effort policy supported by the dataset. Bayesian RL is sometimes recast as belief MDP where the belief is updated upon the new observation, however in our work the dynamics belief is updated to conservatively evaluate policy (we build connection between the belief update and approximate Bayesian inference in Appendix C). It would be interesting to integrate the two types of belief update for the downstream topic, i.e., online adaptation of offline RL.
>    * Bayesian Robust RL [1] considers the robustness by resorting to robust MDP, where the uncertainty set is defined as a L1-norm ball. One contribution there is that the uncertainty set will be updated upon new observations to alleviate the degree of conservativeness. In contrast, our considered AMG is devised to avoid the disadvantages of robust MDP. In this sense, the contributions are orthogonal.
> - CVaR-style algorithms
>    * All of CVaR, robust MDP and our proposed criterion can be deemed as the specializations of Bayesian decision theory, however they are derived from different principles and with different properties. Robust MDP purely focuses on the quantile performance, and ignoring the other possibilities is reported ([25-27] in the paper) to produce over-conservative behavior. CVaR instead considers the average performance of the worst $\delta$-fraction possibilities. Although CVaR involves more information about the stochasticity, it is still solely from the pessimistic view. Recent works propose to improve by maximizing the convex combination of mean performance and CVaR [3], or maximizing mean performance under CVaR constraint [4]. However, they are intractable regarding policy optimization, i.e., proved as an NP-hard problem or relying on heuristic. The AMG formulation presents an alternative way to tackle the entire spectrum of plausible transitions while also give more attention on the pessimistic parts. Besides, the policy optimization is with theoretical guarantee.
>    * Apart from the difference of criterion, [2] also considers the setting of Bayesian RL.
>
> [1] Derman, E., et al. A Bayesian approach to robust reinforcement learning. UAI 2020.\
> [2] Rigter, M., et al. Risk-averse Bayes-adaptive reinforcement learning. NeurIPS 2021.\
> [3] Lobo, E. A., et al. Soft-Robust Algorithms for Batch Reinforcement Learning. arXiv, 2021.\
> [4] Rigter, M., et al. Planning for Risk-Aversion and Expected Value in MDPs. AAAI 2022.

---

### Official Review · Reviewer_LbFj · 2022-07-12

**Rating:** 8
**Confidence:** 5
**Soundness:** 4 excellent
**Presentation:** 3 good
**Contribution:** 4 excellent

**Summary:**

This paper points out that characterizing the impact of dynamics uncertainty through reward penalty may incur unexpected over-conservative behaviors. To overcome this problem in existing model-based offline RL works, the paper proposes to maintain a belief distribution over dynamics belief and optimize policy via biased sampling from the distribution. The sampling procedure is derived based on an AMG formulation of offline RL. For practical consideration, an offline RL approach named PMDB is further designed. Results on the D4RL benchmark show the state-of-the-art performance of PMDB.

**Questions:**

1. I’m not sure if I actually understand the updating of game transitions/dynamics beliefs. Could you please explain it to me in detail?
2. PMDB relies on the initial dynamics belief and the paper uses a uniform distribution over dynamics as the initial belief. Do you mean inserting additional knowledge into the initial dynamics belief is to pre-define an initial dynamic belief? If so, could you give some examples of the scenarios in which we obtain priors on dynamic belief? If involved the expert knowledge, how much will the performance of the algorithm
3. I’m concerned about whether the training of PMDB is stable considering the design of this framework?



**Limitations:**

The paper discussed the limitation that expert knowledge is not always available thus it’s hard to manually pre-define the dynamics belief. Besides, I think another limitation is the lack of ablations on several key components.

**Strengths And Weaknesses:**

Strengths:
* Motivation: The paper points out an important while has been ignored problem in model-based offline RL, that it’s unsuitable to directly characterize the dynamic uncertainty by reward penalty.
* Theoretical Contribution: To solve the problem, the paper formulates the offline RL problem as an AMG and discusses the relationship between AMG and robust MDP. Throughout the formulation, several interesting theoretical results are provided. The illustration that AMG is a successive interpolation between model-based RL and robust MDP provides an intuitive explanation of the theoretical results.
* Empirical Contribution: The paper verifies the proposed PMDB method on D4RL and shows the SOTA performance without the need to tune hyperparameters for each task.

Weaknesses:
* It may be due to the complexity of the overall framework, the paper is somewhat hard to follow. It took me a very long time to understand the symbols and formulations defined in Section 3.1.
* Although the framework is very complicated, it seems the core parts that work are: (1) dynamically adjusting the game transitions in each training iteration; (2) Updating the Q-function based on the K-percentile dynamics instead of the worst one. While with the theoretical proof, it would be better to do ablations on the above two parts to further illustrate the effectiveness of the designs.

---

> ### Author Response · Authors · 2022-08-02
> **Official Response to Reviewer LbFj**
>
> Thanks for your constructive comments. We provide clarification to your questions and concerns as below.
>
> **Q1: The updating of game transitions/dynamics beliefs**
>
> A1: To clarify, let's distinguish the game transition $G$ in AMG and the system transition $T$ in MDP. The game transition $G$ is kept fixed throughout. Theorem 2 states that given an initial belief distribution over $T$, the conservative evaluation produces an updated belief distribution, still over $T$. The belief update is introduced to explain how the AMG connects with MDP, and not truly executed on the algorithmic level. When dealing with policy optimization in Section 4, we do not rely on the equivalent MDP to devise algorithm.
>
> In Theorem 2, the new belief is obtained via reweighting initial belief. The reweighting factor for system transition $\color{blue}\tau^{sa}$ depends on the value of $\mathbb{E}_{{\color{blue}\tau^{sa}},\pi}\big[Q_\{N,k\}^{\pi}(s',a')\big]$. This term can be regarded as a pessimism indicator for $\tau^{sa}$, as it predicts how is the performance if the system transits following $\tau^{sa}$. Note that $\tau^{sa}$ itself is random following the belief distribution, then $\mathbb{E}_\{\tau^{sa},\pi\}\big[Q_\{N,k\}^{\pi}(s',a')\big]$, as a functional of $\tau^{sa}$, is also random. Thus, we can define its cumulative density function, i.e., $F\left(\mathbb{E}_\{ \tau^{sa}, \pi\}\big[Q_\{N, k\}^{\pi}(s',a')\big]\right)$.
> The interesting thing is that the reweighting factor achieves maximum for $\tau^*:F\left(\mathbb{E}_\{\tau^*,\pi\}\big[Q^\pi_\{N,k\}(s',a')\big]\right)=\frac{k-1}{N-1}$, i.e., the transition with $\frac{k-1}{N-1}$-quantile pessimism indicator. Besides, when $\mathbb{E}_\{\tau^{sa}, \pi\}\big[ Q^\pi_\{N,k\}(s',a')\big]$ departs the $\frac{k-1}{N-1}$ quantile, the reweighting coefficient for its $\tau^{sa}$ decreases. In this way, the new belief is reshaped towards concentrating around the $\frac{k-1}{N-1}$ quantile.
>
> **Q2: Inserting additional knowledge, scenarios and expected performance gain.**
>
> A2: The additional knowledge can be inserted by pre-defining more informative initial belief. For example,
> - Consider the physical system where the dynamics can be described as mathematical expression but with uncertain parameter. If we have a narrow distribution over the parameter (according to expert knowledge or inferred from data), the system is almost known for certain. Here, both the mathematical expression and narrow distribution provide more information.
> - Consider the case where we know the dynamics is smooth with probability of 0.7 and periodic with probability of 0.3. Gaussian processes (GPs) with RBF kernel and periodic kernel can well encode these prior knowledges. Then, the 0.7-0.3 mixture of the two GPs trained with offline data can act as the dynamics belief to provide more information.
> - In the case where multi-task datasets are available, we can train dynamics models using each of them and assign likelihood ratios to them. If the likelihood ratio well reflects the similarity between the concerned task and the offline tasks, the multi-task datasets promote knowledge.
>
> The performance gain is expected to monotonously increase with the amount of correct knowledge. As an impractical but intuitive example, with the exact knowledge of system transition (the initial belief is a delta function), the proposed approach is actually optimizing policy as in real system.
>
> **Q3: Stability of training process**
>
> A3: We are not sure whether the concern is due to ''the update of game transition" or the adversarial-style problem formulation. We clarify that the game transition is fixed in A1, the concern could be resolved for the first case. Regarding the general adversarial training, the vanilla gradient-based method does suffer from instability, especially reported in GAN-related works. However, our formulated problem is solved based on the contraction mapping, and it is theoretically guaranteed to converge for fixed reference policy (Theorem 4). For periodically updated reference policy, Theorem 5 gives the condition of monotonous improvement, which is easily satisfied by considering Theorem 4. Our practical algorithm applies slow-evolving reference policy. In the experiment, we keep the hyper-parameter fixed, and do not see unstable behavior regarding online evaluation. We will release code in short future such that it can be verified.
>
> **Ablation experiment**
>
> A4: The core part is formulating AMG to characterize the impact of dynamics uncertainty, but indeed we can do ablation study on the elements of AMG. The impact of whether to choose the worst dynamics is reported in Table 2. Currently, we are doing experiments about how the randomness of candidate set affects, and will post it once done.
>
> **Presentation in Section 3.1**
>
> A5: We will try best to simplify the symbol and formulation in Section 3.1.
>
> **Limitation**
>
> A6: We will add the discussion in A2 to main paper, and include the ablation experiment.

---

> > ### Author Response · Authors · 2022-08-06
> > **Ablation study of the randomness of $\mathcal{T}$**
> >
> > Compared to the standard Bellman backup operator in Q-learning, the proposed one additionally includes the expectation over $\mathcal{T}\sim\mathcal{P}_T^N$ and the $k$-minimum operator over $\tau\in\mathcal{T}$. We report the impact of choosing different $k$ in Table 2, and present the impact of the randomness of $\mathcal{T}$ as below. Fixed $\mathcal{T}$ denotes that we sample $\mathcal{T}$ from the belief distribution and then keep fixed during policy optimization.
> >
> > Task Name| Stochastic $\mathcal{T}$| Fixed $\mathcal{T}$
> > -|-|-
> > hopper-medium        | 106.8$\pm$0.2  | 106.2$\pm$0.3
> > walker2d-medium     | 94.2$\pm$1.1    | 90.1 $\pm$4.3
> > halfcheetah-medium | 75.6$\pm$1.3    |  73.1$\pm$ 2.8
> >
> > We observe that the randomness of $\mathcal{T}$ has a mild effect on the performance in average. The reason can be that we apply the uniform distribution over dynamics ensemble as initial belief (without additional knowledge to insert). The model ensemble is reported to produce low uncertainty estimation in distribution of data coverage and high estimation when departing the dataset [1]. This property makes the optimized policy keep close to the dataset, and it does not rely on the randomness of ensemble elements. However, involving the randomness could lead to more smooth variation of the estimated uncertainty, which benefits the training process and results in better performance. Apart from these empirical results, we highlight that in cases with more informative dynamics belief, only picking several fixed samples from the belief distribution as $\mathcal{T}$ will result in the loss of knowledge.
> >
> > [1] Lakshminarayanan, B., et al. Simple and scalable predictive uncertainty estimation using deep ensembles. NeurIPS 2017.

---

> > > ### Comment · Reviewer_LbFj · 2022-08-07
> > > **Thank you for your effort in writing author responses.**
> > >
> > > I think all my concerns were well addressed in the rebuttal. I will keep my score unchanged. Congratulations to the authors for both the nice emprical and theoretical results!

---

### Official Review · Reviewer_y5WY · 2022-07-12

**Rating:** 7
**Confidence:** 4
**Soundness:** 4 excellent
**Presentation:** 2 fair
**Contribution:** 4 excellent

**Summary:**

The paper proposes an Alternating Markov Game formulation of offline RL which induces a MDP with pessimistic dynamics. The authors derive an approximate solution to this game which requires no uncertainty penalization and show strong results on the D4RL MuJoCo benchmark.

**Questions:**

- How does the slow-evolving policy update relate to TRPO/PPO style KL-constrained policy optimisation?
- It would be interesting to understand more the motivation behind the design of the optimization procedure in Section 4. How inefficient is solving the original problem, what is the overall gain in training speed by using the reference policy?


**Limitations:**

Efficiency of the algorithm should be detailed in the main paper, as well as discussion of related work.

**Strengths And Weaknesses:**

Strengths:
- Strong theoretical background including derivation of connection between AMG and the equivalent MDP, well-related to standard offline MBRL in the $N=k=1$ case.
- Strong empirical results on the D4RL MuJoCo benchmark. The offline evaluation is thorough uses a single hyperparameter setup across all environments, thus avoiding using online samples to tune performance for future deployment.

Weaknesses:
- Presentation is sometimes dense, related work should be discussed in the main paper.

Minor:
- Typo on line 22, 345: trial instead of trail.
- The version of D4RL dataset used in the main evaluation should be highlighted, there are performance differences between v0-v2.
- Section link in line 498 broken.
- [1] considers a related formulation which also induces pessimistic dynamics. It would be valuable to compare these works, but I appreciate this work appeared after the ICML deadline.

[1] Rambo-rl: Robust adversarial model-based offline reinforcement learning. M Rigter, B Lacerda, N Hawes.

---

> ### Author Response · Authors · 2022-08-02
> **Official Response to Reviewer y5WY**
>
> Thanks for your insightful comments. We provide clarification to your questions and concerns as below.
>
> **Q1: How does the slow-evolving policy update related to TRPO/PPO style KL-constrained policy optimisation?**
>
> A1: They both emphasize slow policy update, but come from different derivations and different motivations. As PPO can be regarded as a simplified variant of TRPO, we discuss TRPO in the following.
>
> The slow-evolving policy update produces $\pi_{\phi'}$, where $\phi' \leftarrow \omega_1\phi + (1-\omega_1)\phi'$, and $\pi_{\phi}$ is the policy being optimized to maximize $\bar{J}(\pi_{\phi};\pi_{\phi’})$ in equation (9). Its differences to TRPO regarding derivation are as follows
> - Compared to TRPO which treats $D_\text{KL}(\pi_\text{old}||\pi)\leq \epsilon$ as constraint, $\bar{J}(\pi_{\phi};\pi_{\phi’})$ involves $\mathbb{E}\left[D_\text{KL}(\pi_{\phi}||\pi_{\phi’})\right]$ as regularizer. Note that the expectation is over the state distribution induced by $\pi_{\phi}$ in AMG. Then, during the maximization of $\bar{J}(\pi_{\phi};\pi_{\phi’})$, the regularizer has two effects: keep $\pi_{\phi}$ close to the reference policy; encourage $\pi_{\phi}$ to generate the states where the KL term is small. In this way, $\pi_{\phi}$ is optimized with the consideration of long-term KL regularization. In contrast, the KL constraint in TRPO is not directly affected by the state distribution. This difference about KL regularization/constraint also exists between TRPO and soft actor critic.
> - $\pi_{\phi’}$ is a slow-changing version of $\pi_{\phi}$, while in TRPO $\pi_\text{old}$ is updated as $\pi$ after collecting new data. In our initial experiments, we found the soft update results in faster learning.
>
> Regarding the motivation, TRPO optimizes a first-order approximation to the expected return, and $\pi$ is constrained close to $\pi_\text{old}$ so that the approximation is exact enough to truly improve performance. In our approach, maximizing $\bar{J}(\pi_{\phi};\pi_{\phi’})$ is actually a bi-level problem, where the outer problem is to improve $\pi_\phi$ and the inner problem is to conservatively evaluate policy. Obviously, without sufficient evaluation, the policy improvement will be misled. To avoid this, the KL regularizer restricts $\pi_\phi$ in a small region near $\pi_{\phi'}$, such that these policies can be evaluated sufficiently with limited computation before improvement. The delay between $\pi_\phi$ and $\pi_{\phi'}$ is introduced to provide the time window for sufficiently conservative evaluation.
>
> **Q2: It would be interesting to understand more the motivation behind the design of the optimization procedure in Section 4. How inefficient is solving the original problem, what is the overall gain in training speed by using the reference policy?**
>
> A2: We explain the motivation in A1, and here elaborate why the proposed approach is more efficient than solving the original problem. In general, for the bi-level problem $\max_x \min_y f(x,y)$, a reliable approach is first finding the optimum for inner problem and then taking a single gradient update to the outer variable, i.e., $x\rightarrow x + lr\cdot \nabla_x f(x,y^*)$. In this way, every solved inner problem only contributes a single gradient step to the outer problem. In the proposed approach, we instead constrain the policy in a small region via KL regularizer, and it hopefully produces a more significant policy update after solving each problem regarding $\bar{J}$.
>
> This idea can be compared with that of TRPO. With a batch of new collected data, TRPO applies KL constraint to improve policy more than a single step of vanilla policy gradient. The essential goal is to reduce sample cost. In our setting, the goal is to reduce computational cost. In the initial experiments, we tried alternating policy update (i.e., the policies of primary and secondary players) to solve the original problem, the performance improves stably only with extreme small learning rate for the primary policy, and it is right the inefficiency that motivates us to consider the KL-regularized version.
>
> **Related work: [1] considers a related formulation which also induces pessimistic dynamics. It would be valuable to compare these works, but I appreciate this work appeared after the ICML deadline.**
>
> A3: We will add the full comparison in the paper. Currently, at least for the 12 tasks overlapping with our experiment, the proposed approach achieves an obvious gain (average score: 79.9 with consistent hyper-parameter v.s. 67.7 with tuning hyper-parameter per task). We conject one reason is that the problem formulation in [1] is based on robust MDP.
>
> **Weakness & Limitation**
>
> A4: We will try best to improve the presentation. We will move the related works to main paper, and include the discussion about efficiency (like A2).
>
> **Typos and D4RL detail**
>
> A5: Thanks for pointing them out. We will fix the typos and state the version of D4RL dataset (v2).

---

### Author Response · Authors · 2022-08-09
**Broader Impact**

We thank all reviewers for the constructive and insightful comments, and we are glad that the raised concerns have been resolved. The comments also help us to make the broader impact explicit, which we want to highlight as below:

- **Bayesian Sides of RL:** RL is previously considered from two types of Bayesian view. 1) Bayesian RL [1] applies belief MDP formulation and updates the belief during deployment to promote better decision. The belief is updated upon new observation, while in offline setting no additional data can be received and the proposed belief update comes from our tendency to conservative policy evaluation/optimization. This work can be deemed as an illustration on how to insert subjective preference to the solved policy without relying on data. 2) RL was recast as approximate posterior inference [2], allowing to integrate the policy learning and the reasoning about compositionality and partial observability in a principled manner. In this work, the proposed approach is elaborately recast as EM algorithm with structured posterior inference (in Appendix C). This enables the extension of the related research achievements in online setting to the offline setting, such as hierarchical reinforcement learning [3] and partially observed reinforcement learning [4] based on "RL as inference".

- **Bayesian Decision Theory:** Decision making under uncertainty is a broad and long-standing research topic. Various criterions regarding robustness or conservativeness have been proposed, such as value at risk (VaR) and conditional value at risk (CVaR). In context of RL, robust MDP is a simplified version of VaR. The criterions of VaR and CVaR only focus on the pessimistic performance and ignore the others. In this work, we propose an alternative criterion, which tackles the entire spectrum of plausible transitions while also gives more attention on the pessimistic parts. Although the recent works [5,6] also try to avoid the excessive fixation on pessimism, their formulation is proven NP-hard or the approach relies on heuristic. Apart from RL, we believe this criterion also adapts well to one-step bandit problem (by pessimistically sampling the reward belief).

- **Knowledge Insertion for Data-Limited Policy Optimization:** In offline setting, extra knowledge is strongly desired to further optimize policy. The proposed approach provides an interface to absorb the aforehand knowledge of system transition. On one hand, with richer knowledge, the performance evaluation is more exact and the optimization is away from excessive conservatism. On the other hand, the knowledge can help tuning hyper-parameter with treating the AMG performance as surrogate (as discussed in A2 to Reviewer 998y). However, the knowledge is not easily accessed in general. We hope this work can inspire more research interests on learning data-drivien knowledge from related offline datasets or similar online tasks.

Please kindly let us know if you have any question about these. We thank again the reviewers for the positive comments, and appreciate it if the reviewers think the work deserves higher score.

[1] Ghavamzadeh, M., et al. Bayesian reinforcement learning: A survey. Foundations and Trends in Machine Learning, 2015.\
[2] Levine, S.. Reinforcement learning and control as probabilistic inference: Tutorial and review. arXiv, 2018.\
[3] Haarnoja, T., et al. Latent space policies for hierarchical reinforcement learning. ICML, 2018.\
[4] Lee, Alex X., et al. Stochastic latent actor-critic: Deep reinforcement learning with a latent variable model. NeurIPS, 2020.\
[5] Lobo, E. A., et al. Soft-Robust Algorithms for Batch Reinforcement Learning. arXiv, 2021.\
[6] Rigter, M., et al. Planning for Risk-Aversion and Expected Value in MDPs. AAAI 2022.

---

> ### Comment · Reviewer_LbFj · 2022-08-09
> **To authors**
>
> I have carefully read your new comments. Considering the potential impacts to RL community, I believe this work deserves a higher score. I've raised my score. Congrats to the authors for the remarkable work!

---

> > ### Author Response · Authors · 2022-08-09
> > **To Reviewer LbFj**
> >
> > Thanks very much for your time to further evaluate the value of this work!

---

### Meta-Review · Area_Chair_c4kc · 2022-08-26

**Recommendation:** Accept
**Confidence:** Certain

**Metareview:**

I went through the manuscript, reviews and authors' responses. I think this paper is qualified for NeurIPS publication.

**Award:**

No

---

### Decision · Program_Chairs · 2022-09-14

Accept